# Vg1-Nodal heterodimers are the endogenous inducers of mesendoderm

Tessa G Montague[1]*, Alexander F Schier[1,2,3,4,5]*

[1]Department of Molecular and Cellular Biology, Harvard University, Cambridge, United States; [2]Center for Brain Science, Harvard University, Cambridge, United States; [3]Broad Institute of MIT and Harvard, Cambridge, United States; [4]Harvard Stem Cell Institute, Cambridge, United States; [5]FAS Center for Systems Biology, Harvard University, Cambridge, United States

**Abstract** Nodal is considered the key inducer of mesendoderm in vertebrate embryos and embryonic stem cells. Other TGF-beta-related signals, such as Vg1/Dvr1/Gdf3, have also been implicated in this process but their roles have been unclear or controversial. Here we report that zebrafish embryos without maternally provided *vg1* fail to form endoderm and head and trunk mesoderm, and closely resemble *nodal* loss-of-function mutants. Although Nodal is processed and secreted without Vg1, it requires Vg1 for its endogenous activity. Conversely, Vg1 is unprocessed and resides in the endoplasmic reticulum without Nodal, and is only secreted, processed and active in the presence of Nodal. Co-expression of Nodal and Vg1 results in heterodimer formation and mesendoderm induction. Thus, mesendoderm induction relies on the combination of two TGF-beta-related signals: maternal and ubiquitous Vg1, and zygotic and localized Nodal. Modeling reveals that the pool of maternal Vg1 enables rapid signaling at low concentrations of zygotic Nodal.
DOI: https://doi.org/10.7554/eLife.28183.001

*For correspondence:
tessa.montague@gmail.com
(TGM);
schier@fas.harvard.edu (AFS)

**Competing interests:** The authors declare that no competing interests exist.

## Introduction

The induction of mesoderm and endoderm (mesendoderm) during embryogenesis and embryonic stem cell differentiation generates the precursors of the heart, liver, gut, pancreas, kidney and other internal organs. Nodal, a ligand in the TGF-beta protein family, is the key inducer of vertebrate mesendoderm (*Schier and Shen, 2000*; *Schier, 2009*; *Shen, 2007*), ranging from zebrafish and mouse embryos to human embryonic stem cells. Nodal mutants fail to form mesendodermal cell lineages in zebrafish and mouse (*Conlon et al., 1991*; *1994*; *Feldman et al., 1998*; *Zhou et al., 1993*), and activation of the Nodal signaling pathway drives the *in vitro* differentiation of embryonic stem cells into mesendodermal progenitors (*Brandenberger et al., 2004*; *Camus et al., 2006*; *D'Amour et al., 2005*; *Hoveizi et al., 2014*; *Kubo et al., 2004*; *Parisi et al., 2003*; *Schier and Shen, 2000*; *Shen, 2007*; *Smith et al., 2008*; *Takenaga et al., 2007*; *Vallier et al., 2004*; *Yasunaga et al., 2005*). Following its role in mesendoderm induction, Nodal activity also patterns the left-right axis. Nodal ligands are expressed in the left lateral plate mesoderm (*Collignon et al., 1996*; *Levin et al., 1995*; *Long et al., 2003*; *Lowe et al., 1996*; *Pagán-Westphal and Tabin, 1998*), and mutants that lack left-sided Nodal signaling exhibit multiple left-right defects (*Brennan et al., 2002*; *Kumar et al., 2008*; *Long et al., 2003*; *Noël et al., 2013*; *Saijoh et al., 2003*; *Yan et al., 1999*).

Nodal is not the only TGF-beta-related signal implicated in mesendoderm induction and left-right patterning. Members of the Vg1/GDF1/GDF3 TGF-beta subfamily have been assigned various roles in these processes, although there are puzzling contradictions from the level of gene expression to the loss-of-function and gain-of-function phenotypes. The role of GDF1 in left-right patterning is well established. *Gdf1* mutant mice exhibit left-right asymmetry defects (*Rankin et al., 2000*) and morpholino studies indicate that zebrafish *vg1* (*dvr1/gdf3*) is required for left-right patterning

**eLife digest** All animals begin life as just one cell – a fertilized egg. In order to make a recognizable adult, each embryo needs to make the three types of tissue that will eventually form all of the organs: endoderm, which will form the internal organs; mesoderm, which will form the muscle and bones; and ectoderm, which will generate the skin and nervous system.

All vertebrates – animals with backbones like fish and humans – use the so-called Nodal signaling pathway to make the endoderm and mesoderm. Nodal is a signaling molecule that binds to receptors on the surface of cells. If Nodal binds to a receptor on a cell, it instructs that cell to become endoderm or mesoderm. As such, Nodal is critical for vertebrate life. However, there has been a 30-year debate in the field of developmental biology about whether a protein called Vg1, which has a similar molecular structure as Nodal, plays a role in the early development of vertebrates.

Zebrafish are often used to study animal development, and Montague and Schier decided to test whether these fish need the gene for Vg1 (also known as Gdf3) by deleting it using a genome editing technique called CRISPR/Cas9. It turns out that female zebrafish can survive without this gene. Yet, when the offspring of these females do not inherit the instructions to make Vg1 from their mothers, they fail to form the endoderm and mesoderm. This means that the embryos do not have hearts, blood or other internal organs, and they die within three days. Two other groups of researchers have independently reported similar results.

The findings reveal that Vg1 is critical for the Nodal signaling pathway to work in zebrafish. Montague and Schier then showed that, in this pathway, Nodal does not activate its receptors on its own. Instead, Nodal must interact with Vg1, and it is this Nodal-Vg1 complex that activates receptors, and instructs cells to become endoderm and mesoderm.

Scientists currently use the Nodal signaling pathway to induce human embryonic stem cells growing in the laboratory to become mesoderm and endoderm. As such, these new findings could ultimately help researchers to grow tissues and organs for human patients.

DOI: https://doi.org/10.7554/eLife.28183.002

(*Peterson et al., 2013*). GDF1/Vg1 alone is unable to activate the Nodal signaling pathway, but it increases the activity and range of mouse and zebrafish Nodal ligands in *Xenopus* assays (*Peterson et al., 2013*; *Tanaka et al., 2007*) and the activity of mouse Nodal in tissue culture cells (*Andersson et al., 2007*; *Fuerer et al., 2014*). Thus, Nodal and Vg1/GDF1 family members cooperate to pattern the left-right axis.

The role of the Vg1/GDF1/GDF3 TGF-beta subfamily in mesendoderm formation is less clear. In *Xenopus* – where Vg1 was first discovered – *vg1* mRNA is localized to a vegetal crescent in the oocyte and in the vegetal hemisphere of the early embryo (*Rebagliati et al., 1985*; *Weeks and Melton, 1987*). By contrast, zebrafish *vg1* mRNA is localized to the animal pole of late stage oocytes (*Marlow and Mullins, 2008*), and it is present ubiquitously in the early embryo (*Dohrmann et al., 1996*; *Helde and Grunwald, 1993*; *Peterson et al., 2013*). *Gdf1* and *Gdf3*, which are considered to be the mammalian Vg1 orthologs (*Andersson et al., 2007*; *Chen et al., 2006*; *Rankin et al., 2000*; *Wall et al., 2000*), are expressed in the 16-cell morula (*Gdf3*) and epiblast prior to gastrulation (*Gdf3 and Gdf1*) (*Chen et al., 2006*; *Wall et al., 2000*). Some *Gdf3* mutants lack a subset of endodermal and mesodermal markers, while others grow to fertile adults (*Andersson et al., 2007*; *Chen et al., 2006*); conversely *Gdf1* mutants only exhibit left-right asymmetry defects (*Rankin et al., 2000*). Some *Gdf1;Gdf3* double mutants exhibit more severe defects in endoderm and mesoderm formation than *Gdf3* single mutants (*Andersson et al., 2007*), and *Gdf1-/-;Nodal+/-*mutants resemble hypomorphic *nodal* mutants (*Andersson et al., 2006*; *Lowe et al., 2001*), suggesting some synergy between GDF1 and Nodal functions. Experiments in the chick and mouse indicate that Vg1/GDF1/GDF3 may act upstream of Nodal (*Andersson et al., 2007*; *Chen et al., 2006*; *Rankin et al., 2000*; *Shah et al., 1997*; *Skromne and Stern, 2001*; *Tanaka et al., 2007*). Thus, these analyses suggest that mouse Nodal, GDF1 and GDF3 may cooperate during early amniote development, but their regulatory and molecular relationships have remained unclear. Morphant studies in zebrafish suggest a function for Vg1 in left-right axis formation but not in mesendoderm induction

(*Peterson et al., 2013*). Antisense oligonucleotide-mediated knockdown of *Xenopus vg1* leads to defects in dorsal mesoderm induction (*Birsoy et al., 2006*), but most mesendodermal derivatives still form. Taken together, loss-of-function studies establish crucial roles for both Nodal and Vg1/GDF1 in left-right development and for Nodal in mesendoderm induction, but the roles of GDF1/GDF3/Vg1 in mesendoderm induction remain poorly understood.

Another puzzling aspect of Vg1's function is its apparent inability to be processed and secreted. This is in stark contrast to other members of the TGF-beta superfamily, which are generated as pro-proteins that dimerize and are cleaved to generate a secreted, mature dimer that binds receptors (*Constam, 2014*; *Dutko and Mullins, 2011*). Neither cleavage nor secretion of Vg1 has been detected in *Xenopus* and zebrafish, and correspondingly, overexpression does not yield a phenotype (*Dale et al., 1993*; *Dale et al., 1989*; *Dohrmann et al., 1996*; *Tannahill and Melton, 1989*; *Thomsen and Melton, 1993*). Conflicting results have been reported for GDF1 processing in heterologous systems, ranging from cleavage but inactivity in *Xenopus* oocytes (*Tanaka et al., 2007*) to no detectable cleavage in *Xenopus* embryos (*Wall et al., 2000*). Mouse Nodal-GDF1 heterodimers, but not zebrafish Nodal-Vg1 heterodimers, have been detected in a heterologous *Xenopus* system (*Peterson et al., 2013*; *Tanaka et al., 2007*). Upon fusion of the Vg1, GDF1 or GDF3 mature domain to the Activin or BMP prodomain, Vg1 is processed and induces mesoderm formation (*Chen et al., 2006*; *Dale et al., 1993*; *Dohrmann et al., 1996*; *Kessler and Melton, 1995*; *Thomsen and Melton, 1993*; *Wall et al., 2000*). However, it is unclear if these constructs reveal the true nature of Vg1, or whether the fused prodomains generate ectopic functions. Thus, it remains to be resolved how Vg1 processing, secretion, dimerization and activity are regulated.

In this study we address the long-standing question of Vg1's role in vertebrate mesendoderm induction and its relationship to Nodal, using zebrafish as a model system. Current models of zebrafish mesendoderm induction have focused entirely on the roles of the two zebrafish Nodal genes, *cyclops* (*cyc*) and *squint* (*sqt*), with no consideration of *vg1* (*Bodenstine et al., 2016*; *Cartwright et al., 2008*; *Chea et al., 2005*; *Constam, 2009*; *Hirokawa et al., 2006*; *Juan and Hamada, 2001*; *Liang and Rubinstein, 2003*; *Papanayotou and Collignon, 2014*; *Pauklin and Vallier, 2015*; *Quail et al., 2013*; *Robertson, 2014*; *Schier and Shen, 2000*; *Shen, 2007*; *Signore et al., 2016*; *Strizzi et al., 2012*; *2009*; *Tian and Meng, 2006*; *Wang and Tsang, 2007*; *Whitman, 2001*). *cyc* and *sqt* are zygotically-expressed at the embryonic margin and act as concentration-dependent inducers of mesendoderm (*Schier, 2009*). *cyc;sqt* double mutants (*Feldman et al., 1998*) and other zebrafish Nodal signaling mutants (*Dubrulle et al., 2015*; *Gritsman et al., 1999*) fail to form endoderm and head and trunk mesoderm. Conversely, ectopic expression of *cyc* or *sqt* induces mesendoderm formation (*Bisgrove et al., 1999*; *Feldman et al., 1998*; *Gritsman et al., 1999*; *2000*; *Meno et al., 1999*; *Sampath et al., 1998*). These results, and the lack of a *vg1* morphant mesendoderm phenotype (*Peterson et al., 2013*), have been interpreted to mean that Cyc and Sqt are the sole inducers of mesendoderm, without a requirement for Vg1 or other TGF-beta family members. Contrary to these models, we now report that *vg1* is absolutely essential for mesendoderm induction. Vg1 is only secreted, processed and active in the presence of Nodal, while Nodal requires Vg1 for activity. Co-expression of Nodal and Vg1 results in heterodimer formation and mesendoderm induction.

## Results

### Maternal *vg1* is required for mesendoderm formation

To determine the function of zebrafish Vg1, we generated *vg1* mutants using CRISPR/Cas9 (*Figure 1—figure supplement 1A*). We recovered 8 bp and 29 bp deletion alleles that cause frameshifts, truncating Vg1 from a 355 amino acid protein to predicted 18 and 11 amino acid peptides, respectively (*Figure 1—figure supplement 1B*). Zygotic homozygous *vg1* (Z*vg1*) mutants were viable, with no strong left-right asymmetry defects (*Figure 1—figure supplement 2*), allowing the generation of maternal *vg1* (M*vg1*) mutants from homozygous females crossed to wild-type males (*Figure 1A*). M*vg1* embryos lacked the derivatives of the mesendoderm, including heart, blood, pronephros, notochord, gut and trunk somites (*Figure 1A*). To test whether the phenotype is caused by the loss of *vg1*, we performed rescue experiments by injecting 5 concentrations of *vg1* mRNA, spanning a 1600-fold range. 0.5–100 pg of *vg1* rescued the phenotype, revealing that the embryo can

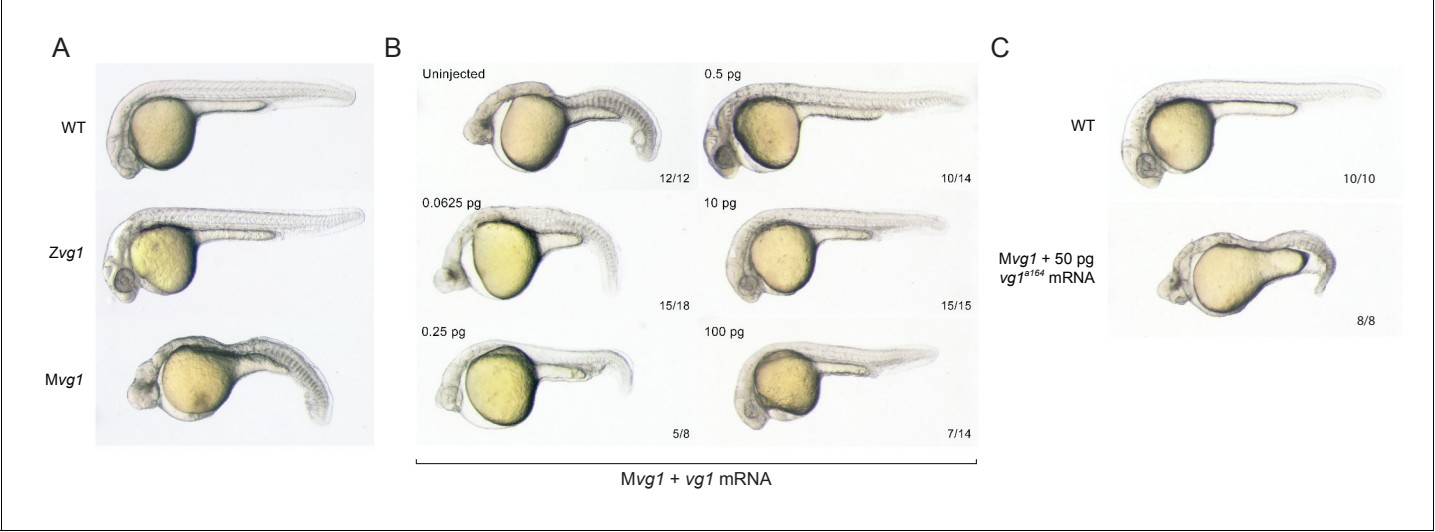

**Figure 1.** Maternal *vg1* is required for mesendoderm formation. (**A**) Zygotic and maternal *vg1* (Z*vg1* and M*vg1*) mutants and wild-type (WT) embryo at 28 hours post-fertilization (hpf). See *Figure 1—figure supplement 1* for information about the *vg1* mutant alleles, and *Figure 1—figure supplement 2* for analysis of left-right asymmetry in WT and Z*vg1* embryos. (**B**) M*vg1* embryos injected with 0.0625–100 pg of *vg1* mRNA. (**C**) M*vg1* embryo injected with 50 pg of *vg1* mRNA containing the 8 bp deletion found in the genetic mutants (*vg1ᵃ¹⁶⁴*).
DOI: https://doi.org/10.7554/eLife.28183.003

The following figure supplements are available for figure 1:

**Figure supplement 1.** *vg1* mutant alleles.
DOI: https://doi.org/10.7554/eLife.28183.004
**Figure supplement 2.** Left-right asymmetry in Z*vg1* mutants.
DOI: https://doi.org/10.7554/eLife.28183.005

tolerate a large range of *vg1* concentrations (*Figure 1B*). 50 pg of a *vg1* mRNA containing the 8 bp deletion found in the genetic mutant was unable to rescue the phenotype (*Figure 1C*). In contrast to previous *vg1* morpholino experiments (*Peterson et al., 2013*), these results reveal that *vg1* is essential for mesendoderm formation.

## Endogenous Nodal signaling requires Vg1

The phenotype of M*vg1* and maternal-zygotic *vg1* (MZ*vg1)* embryos closely resembles that of embryos that lack Nodal (*Feldman et al., 1998*), its co-receptor Oep (*Gritsman et al., 1999*), or its signal transducer Smad2 (*Dubrulle et al., 2015*) (*Figure 2A*). To determine whether M*vg1* embryos are defective in Nodal signaling, we analyzed the expression of a selection of Nodal target genes. The expression of these mesendoderm genes showed the same defects in M*vg1* mutants as in Nodal signaling mutants, indicating that Nodal signaling is not functional in the absence of Vg1 (*Figure 2B*).

One way Nodal signaling might be disrupted in M*vg1* embryos is through loss of Nodal gene expression. We analyzed *cyc* and *sqt* expression in wild-type and M*vg1* embryos. *cyc* and *sqt* were initially expressed at comparable levels across both genotypes, but mRNA levels subsequently increased in wild-type embryos by autoregulation (*Meno et al., 1999*) while they generally remained low in M*vg1* embryos (*Figure 2C*). These results suggest that Vg1 is required for the auto-induction but not initiation of Nodal gene expression, and that the remaining endogenous levels of Nodal are not able to induce mesendoderm in the absence of Vg1.

To test whether the Nodal ligands might be inactive in the absence of Vg1, we overexpressed *cyc* or *sqt* in M*vg1* embryos and analyzed Nodal target gene expression. High levels (50 pg of mRNA) of *cyc* failed to induce target gene expression in M*vg1* embryos (*Figure 2D*), whereas *sqt* at low (0.2 pg) but not high (2–50 pg) levels of overexpression failed to induce target gene expression (*Figures 2D and E*). We then co-expressed 20 pg of *cyc* mRNA with increasing concentrations of *vg1* mRNA in M*vg1* embryos. Co-expression of *cyc* and 5 pg of *vg1* caused an increase in

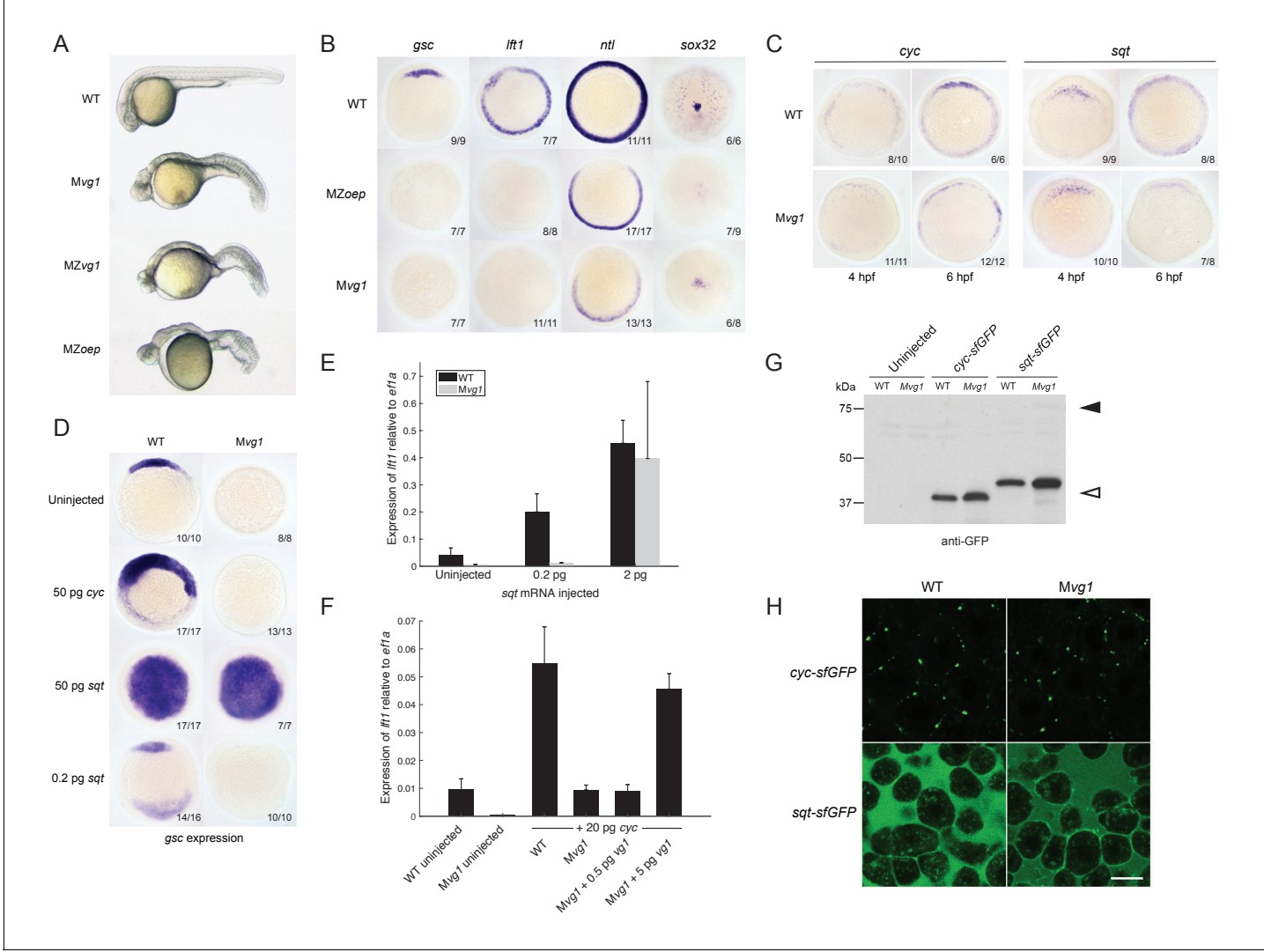

**Figure 2.** Endogenous Nodal signaling requires Vg1. (A) M*vg1*, maternal-zygotic *vg1* (MZ*vg1*) and maternal-zygotic *oep* (MZ*oep*) mutants at 28 hpf. (B) Expression of Nodal target genes *gsc*, *lft1* and *ntl* at 50% epiboly and *sox32* at 90% epiboly in WT, M*vg1* and MZ*oep* embryos. (C) *cyc* and *sqt* expression at 4 and 6 hpf in WT and M*vg1* embryos. (D) *gsc* expression at 50% epiboly in WT and M*vg1* embryos injected with 50 pg of *cyc* mRNA, 50 pg or 0.2 pg of *sqt* mRNA. (E) qPCR of *lft1* expression at 50% epiboly relative to *ef1a* in WT and M*vg1* embryos injected with 0.2 pg or 2 pg of *sqt* mRNA. The mean and standard error of the mean (SEM) was plotted. (F) qPCR of *lft1* expression at 50% epiboly relative to *ef1a* in embryos injected with 20 pg of *cyc* mRNA in combination with 0.5 pg or 5 pg of *vg1* mRNA. The mean and SEM was plotted. (G) Anti-GFP reducing immunoblot of WT and M*vg1* embryos injected with 50 pg of *cyc-sfGFP* or *sqt-sfGFP* mRNA. Black arrowhead indicates the position of full-length protein, open arrowhead indicates processed protein. 8 embryos at 50% epiboly were loaded per well. (H) Live imaging of the animal cap of sphere-stage WT and M*vg1* embryos injected with 50 pg of *cyc-sfGFP* or *sqt-sfGFP* mRNA. Scale bar, 17 um.

DOI: https://doi.org/10.7554/eLife.28183.006

The following source data is available for figure 2:

**Source data 1.** Raw qPCR data for *Figure 2E and F*.
DOI: https://doi.org/10.7554/eLife.28183.007

induction of Nodal target gene expression compared to *cyc* alone (*Figure 2F*). These results indicate that Vg1 is necessary for Cyc activity and partially needed for Sqt activity.

To determine whether the Nodal ligands are processed and secreted in M*vg1* embryos, we expressed superfolderGFP (sfGFP)-tagged derivatives of Cyc and Sqt (*Müller et al., 2012*; *Pédelacq et al., 2006*). No differences in cleavage or localization of Cyc and Sqt were detected in

the presence or absence of Vg1 (*Figures 2G and H*). Taken together, these results suggest that Vg1 is necessary for the endogenous activities, but not the processing and secretion, of Cyc and Sqt.

## Vg1 processing requires Nodal

Previous studies did not detect Vg1 processing in early embryos (*Dale et al., 1989*; *Dohrmann et al., 1996*; *Tannahill and Melton, 1989*; *Thomsen and Melton, 1993*). To examine the relationship of Vg1 processing to presence or absence of Nodal proteins, we first inserted sfGFP downstream of the Vg1 cleavage site (*Figure 3A*). *vg1-sfGFP* rescued M*vg1* mutants (*Figure 3B*) but cleavage of Vg1 protein was undetectable (*Figure 3C*). To test whether Vg1 needs to be processed to be functional, we mutated the basic residues in the Vg1 cleavage site to non-basic residues (Vg1-NC, 'Non-Cleavable' (*Figure 3E*)). This abolished Vg1 rescuing activity (*Figure 3D*), suggesting that endogenous Vg1 cleavage is not detectable but is required for Vg1 function.

Given that the M*vg1* phenotype resembles Nodal loss-of-function phenotypes, and Vg1 requires its cleavage site, we asked whether Nodal might induce Vg1 cleavage. We co-expressed *vg1-sfGFP* with *cyc* or *sqt* and discovered that Vg1-sfGFP was cleaved to its mature form in the presence of Nodal (*Figure 3E*, *Figure 3—figure supplement 1*). By contrast, Vg1-sfGFP was not cleaved upon co-expression with an alternative TGF-beta-related ligand, *bmp7a* (*Figure 3—figure supplement 1B*), and non-cleavable Vg1-sfGFP (Vg1-NC-sfGFP) was not cleaved in the presence of Cyc or Sqt (*Figure 3E*, *Figure 3—figure supplement 1A*). These data reveal that Nodal induces Vg1 processing.

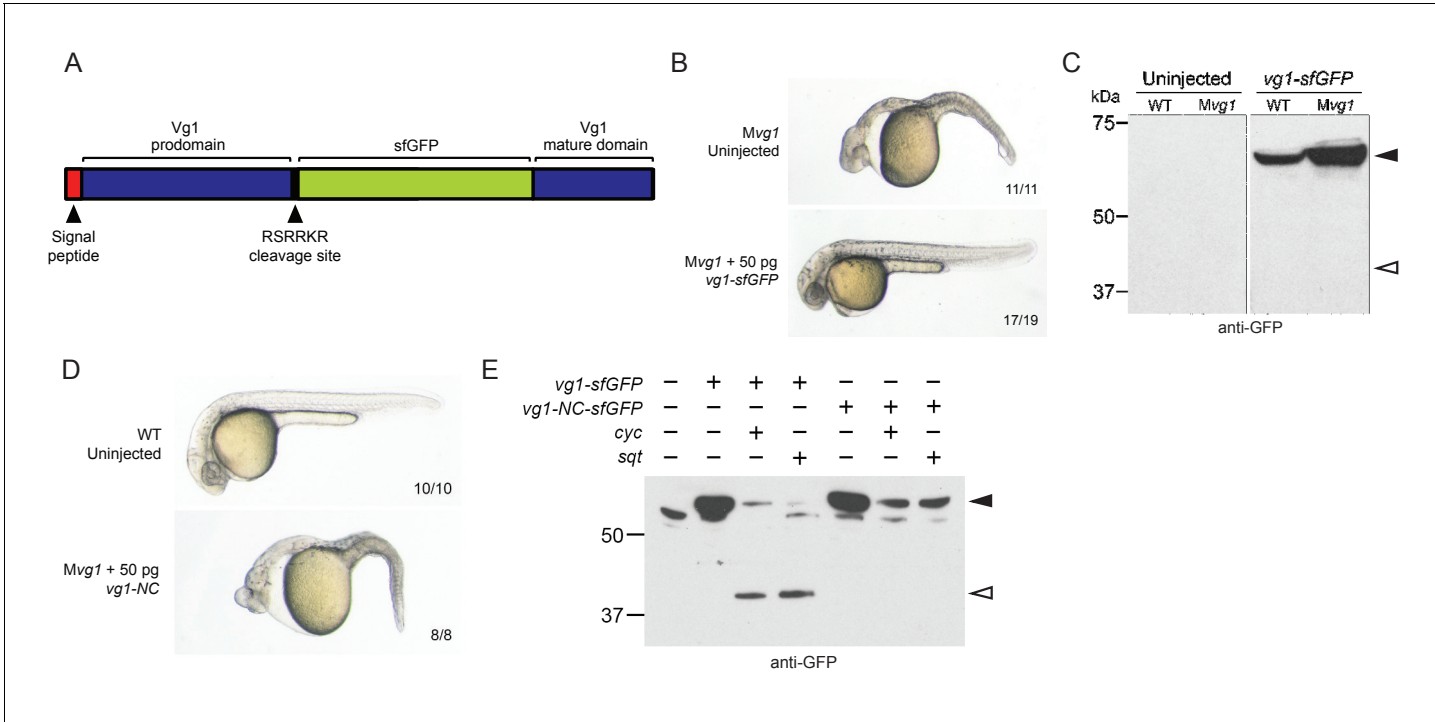

**Figure 3.** Vg1 processing requires Nodal. (**A**) *superfolderGFP* (*sfGFP*) was inserted into *vg1* downstream of the predicted basic cleavage site. (**B**) M*vg1* embryo injected with 50 pg of *vg1-sfGFP* mRNA, shown at 28 hpf. (**C**) Anti-GFP reducing immunoblot of WT and M*vg1* embryos injected with 50 pg of *vg1-sfGFP* mRNA. Black arrowhead indicates full-length Vg1-sfGFP; open arrowhead indicates the predicted size of cleaved Vg1-sfGFP. 8 embryos at 50% epiboly were loaded per well. (**D**) M*vg1* embryo injected with 50 pg of non-cleavable *vg1* mRNA (*vg1-NC*, RSRRKR->SQNTSN), shown at 28 hpf. Embryos were injected with up to 200 pg of *vg1-NC* mRNA with no rescue. (**E**) Anti-GFP reducing immunoblot of M*vg1* embryos injected with 10 pg of *vg1-sfGFP* or *vg1-NC-sfGFP* mRNA and 10 pg of *cyc* or *sqt* mRNA. Black arrowhead indicates full-length Vg1-sfGFP, open arrowhead indicates cleaved Vg1-sfGFP. Molecular weights in kDa. 8 embryos at 50% epiboly were loaded per well. See also *Figure 3—figure supplement 1*.

DOI: https://doi.org/10.7554/eLife.28183.008

The following figure supplement is available for figure 3:

**Figure supplement 1.** Vg1 processing requires Nodal.
DOI: https://doi.org/10.7554/eLife.28183.009

# Vg1 secretion requires Nodal

To examine the secretion and localization of Vg1, we expressed *vg1-sfGFP* in wild-type or M*vg1* embryos for *in vivo* imaging. In contrast to the extracellular localization of Cyc and Sqt (*Figure 2H*),

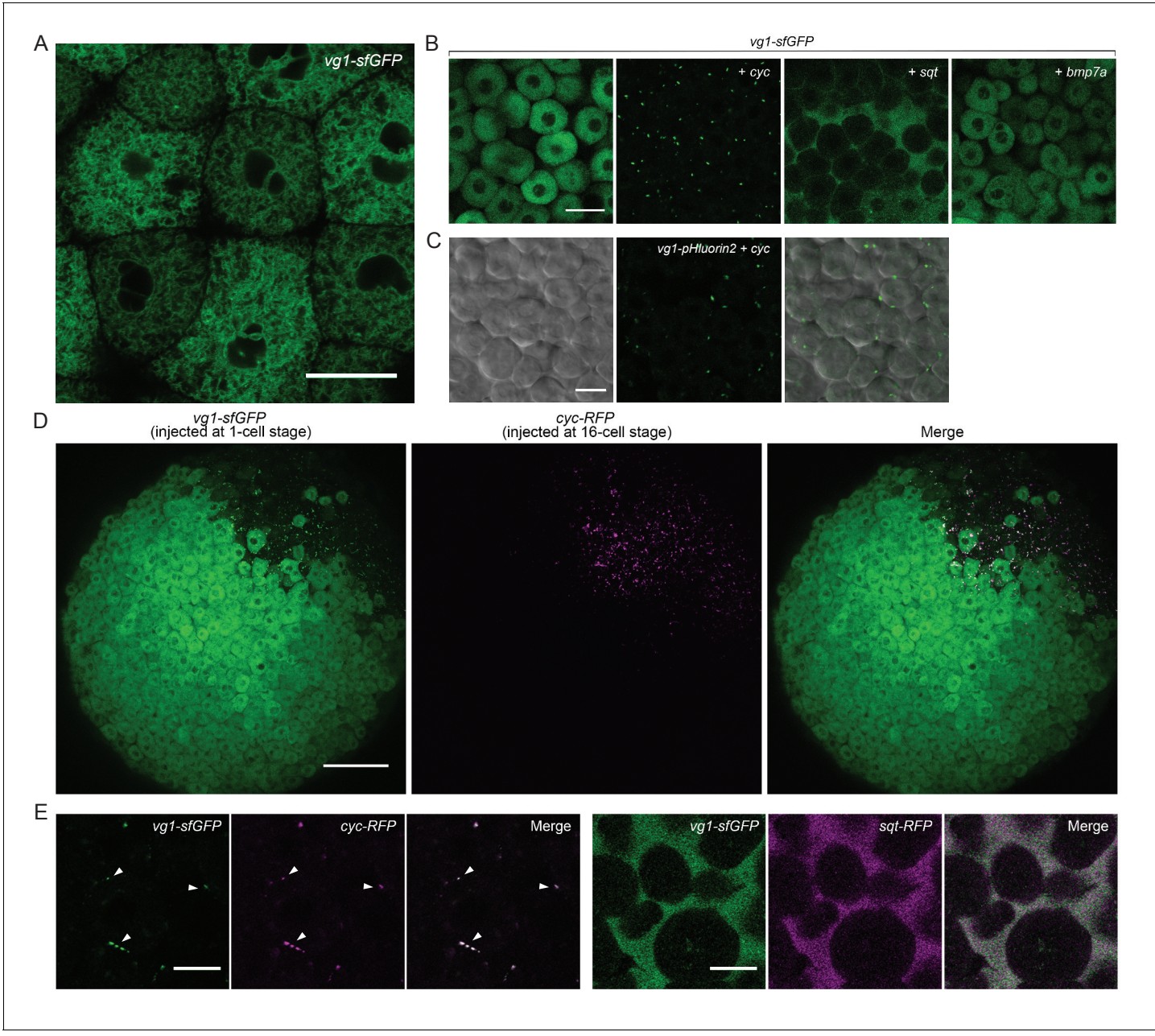

**Figure 4.** Vg1 secretion requires Nodal. (**A**) Live imaging of M*vg1* embryo injected with 50 pg of *vg1-sfGFP* mRNA. Scale bar, 17 um. (**B**) M*vg1* embryos co-injected with 50 pg of *vg1-sfGFP* mRNA and 50 pg of *cyc, sqt* or *bmp7a* mRNA. Scale bar, 17 um. See also *Figure 4—figure supplement 1A*. (**C**) M*vg1* embryo co-injected with 50 pg of pH-sensitive fluorescent *vg1* (*vg1-pHluorin2*) and 50 pg of *cyc* mRNA. Scale bar, 17 um. See also *Figure 4— figure supplement 1B*. (**D**) M*vg1* embryo co-injected with 50 pg of *vg1-sfGFP* mRNA at the 1-cell stage and 10 pg of *cyc-RFP* mRNA into 1 cell at the 16-cell stage. Scale bar, 100 um. (**E**) M*vg1* embryos co-injected with 50 pg of *vg1-sfGFP* mRNA and 50 pg of *cyc-* or *sqt-RFP* mRNA. Arrowheads indicate examples of co-localization. Scale bar, 17 um. See also *Figure 4—figure supplement 1C*.

DOI: https://doi.org/10.7554/eLife.28183.010

The following figure supplement is available for figure 4:

**Figure supplement 1.** Vg1 secretion requires Nodal.
DOI: https://doi.org/10.7554/eLife.28183.011

Vg1 was only detected intracellularly, predominantly in the endoplasmic reticulum (ER) (*Figure 4A*) (*Fodero-Tavoletti et al., 2005*; *Southall et al., 2006*; *Szul and Sztul, 2011*; *Tu et al., 2002*).

To determine whether Nodal can induce not only Vg1 processing but also secretion, we co-expressed *vg1-sfGFP* with *cyc* or *sqt*. Notably, Vg1-sfGFP formed extracellular puncta and/or diffuse extracellular signal upon co-expression with *cyc* or *sqt* (*Figure 4B*, *Figure 4—figure supplement 1A*, *Table 1*). By contrast, Vg1-sfGFP was not secreted upon co-expression with *bmp7a* (*Figure 4B*).

To directly test whether Vg1 is secreted in the presence of Nodal, we tagged Vg1 with the pH-sensitive fluorescent protein pHluorin2 (*Mahon, 2011*). pHluorin2 is non-fluorescent at acidic pH, as found in intracellular vesicles, but it fluoresces in the neutral pH of the extracellular space. Vg1-pHluorin2 fluorescent puncta were only visible upon co-expression with *cyc* or *sqt*, indicating that Vg1 is secreted in the presence of Nodal (*Figure 4C*, *Figure 4—figure supplement 1B*).

To independently test if Vg1 is only secreted in the presence of Nodal, we expressed *vg1-sfGFP* in single-cell embryos and co-expressed *cyc-RFP* in 1 cell at the 16-cell stage. At sphere stage, Vg1-sfGFP was only secreted in the cells that also expressed Cyc-RFP (*Figure 4D*).

To determine whether Vg1 and Nodal co-localize, we co-expressed *vg1-sfGFP* with *cyc-RFP* or *sqt-RFP*. Vg1-sfGFP displayed extensive extracellular co-localization with Cyc-RFP and Sqt-RFP (*Figure 4E*, *Figure 4—figure supplement 1C*). Taken together, these results reveal that Nodal induces the secretion of Vg1, and that Vg1 and Nodal co-localize.

## Vg1 and Nodal form heterodimers

The co-localization of Vg1 and Nodal suggested that these secreted ligands might form heterodimers, as detected for GDF1 and Nodal, and some other TGF-beta-related signals (*Aono et al., 1995*; *Dutko and Mullins, 2011*; *Eimon and Harland, 2002*; *Fuerer et al., 2014*; *Guo and Wu, 2012*; *Hazama et al., 1995*; *Israel et al., 1996*; *Little and Mullins, 2009*; *Nishimatsu and Thomsen, 1998*; *Schmid et al., 2000*; *Shimmi et al., 2005*; *Suzuki et al., 1997*; *Tanaka et al., 2007*). To test this hypothesis, we performed co-immunoprecipitation experiments by co-expressing 50 pg of *vg1-Flag* with 50 pg of *cyc-HA* or *sqt-HA*. Vg1-Flag co-immunoprecipitated with Cyc-HA or Sqt-HA (*Figure 5A*, *Figure 5—figure supplement 1A*). To test the specificity of this interaction we used two different concentrations of *sqt-HA* mRNA in combination with three different concentrations of *vg1-Flag* or *bmp7a-Flag* mRNA. We detected an interaction between Sqt-HA and Vg1-Flag at all six concentrations tested, whereas an interaction between Sqt-HA and Bmp7a-Flag was only detected at

**Table 1.** Quantification of Vg1-sfGFP localization in M*vg1* embryos co-injected with 20 pg of *vg1-sfGFP* mRNA and 0.5–20 pg of *cyc* or *sqt* mRNA.
See *Figure 4—figure supplement 1A* for examples of Vg1-sfGFP extracellular localization.

| mRNA co-injected with 20 pg *vg1-sfGFP* | | Intracellular | Extracellular puncta | Extracellular diffuse | Extracellular puncta + diffuse |
|---|---|---|---|---|---|
| *cyc* (pg) | 0.5 | 2 | 3 | 0 | 0 |
| | 1 | 5 | 5 | 0 | 0 |
| | 2 | 0 | 8 | 0 | 0 |
| | 5 | 0 | 7 | 0 | 0 |
| | 10 | 0 | 6 | 0 | 0 |
| | 20 | 0 | 9 | 0 | 0 |
| *sqt* (pg) | 0.5 | 5 | 0 | 0 | 0 |
| | 1 | 4 | 4 | 1 | 0 |
| | 2 | 0 | 3 | 2 | 4 |
| | 5 | 0 | 6 | 0 | 0 |
| | 10 | 2 | 0 | 3 | 5 |
| | 20 | 0 | 2 | 5 | 3 |

DOI: https://doi.org/10.7554/eLife.28183.012

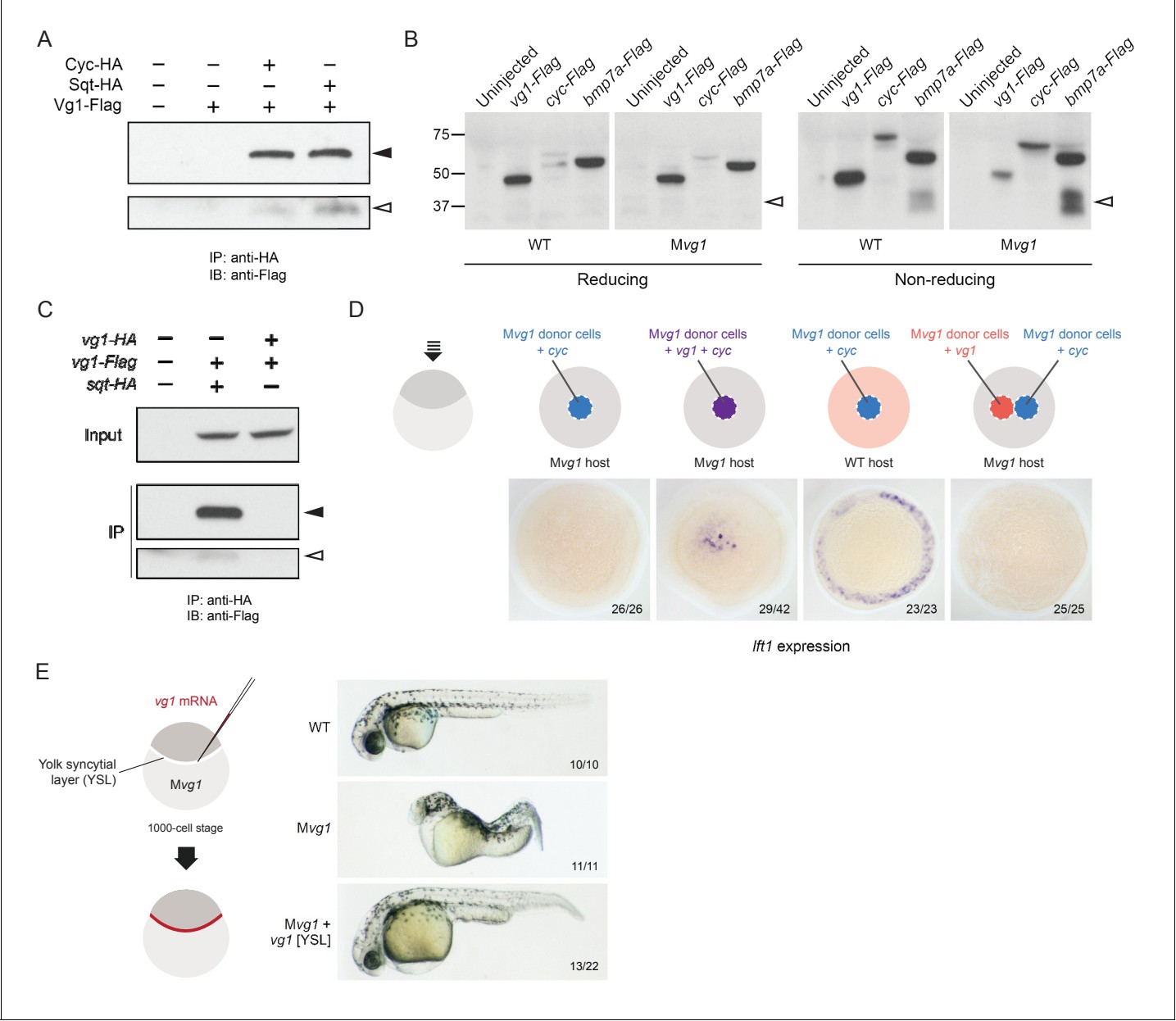

**Figure 5.** Vg1 and Nodal form heterodimers, and are only active when co-expressed. (**A**) Anti-Flag reducing immunoblot (IB) of anti-HA immunoprecipitates (IP) from lysates of M*vg1* embryos injected with 50 pg of *cyc-HA*, *sqt-HA* and/or 50 pg of *vg1-Flag* mRNA. Black arrowhead indicates full-length Vg1-Flag; open arrowhead indicates cleaved Vg1-Flag. See also *Figure 5—figure supplements 1A and B*. (**B**) Anti-Flag reducing and non-reducing immunoblots of WT and M*vg1* embryos injected with 50 pg of *vg1-Flag*, *cyc-Flag* or *bmp7a-Flag*, collected at 50% epiboly. Under reducing conditions the proteins migrated at sizes consistent with the theoretical molecular weights for full-length monomers: Vg1-Flag – 42 kDa; Cyc-Flag – 58 kDa; Bmp7a-Flag – 50 kDa. Open arrowhead indicates where mature Bmp7a-Flag homodimers are expected to migrate as two species under non-reducing conditions (*Little and Mullins, 2009*). For an annotated gel, see *Figure 5—figure supplement 1C*. (**C**) Anti-Flag reducing immunoblot of anti-HA IP from lysates of M*vg1* embryos injected with 50 pg of *sqt-HA* and *vg1-Flag* mRNAs or *vg1-HA* and *vg1-Flag* mRNAs. Black arrowhead indicates full-length Vg1-Flag; open arrowhead indicates cleaved Vg1-Flag. (**D**) Transplantation of cells from donor embryos injected with *cyc*, *vg1* or *cyc* and *vg1* mRNAs into host embryos for analysis of Nodal target gene *lft1* expression at 50% epiboly. mRNAs were co-injected with *sfGFP* mRNA to verify successful transplantation using DAB staining (*Figure 5—figure supplement 2A*). (**E**) Injection of *vg1* mRNA into the yolk syncytial layer (YSL) of M*vg1* mutants, shown at 32 hpf. For Nodal target gene expression in *vg1* mRNA YSL-injected embryos see *Figure 5—figure supplement 2B*. *vg1* was co-injected with a fluorescent dextran to verify YSL localization (*Figure 5—figure supplement 2C*).

DOI: https://doi.org/10.7554/eLife.28183.013

The following figure supplements are available for figure 5:

**Figure supplement 1.** Vg1 and Nodal form heterodimers.

*Figure 5 continued on next page*

*Figure 5 continued*

DOI: https://doi.org/10.7554/eLife.28183.014

**Figure supplement 2.** Vg1 and Nodal are only active when co-expressed.

DOI: https://doi.org/10.7554/eLife.28183.015

the highest concentration of each mRNA (*Figure 5—figure supplement 1B*). Thus, Vg1 specifically interacts with Nodal to form heterodimers.

The heterodimerization of Vg1 and Nodal raises the possibility that Vg1 is maintained in a monomeric state in the absence of Nodal. Indeed, a previous study found that Vg1 does not form homodimers, and that endogenous Vg1 is predominantly monomeric (*Dale et al., 1993*). To test the monomeric or dimeric states of Vg1 in the absence of Nodal, we performed reducing and non-reducing immunoblots of wild-type and M*vg1* embryos expressing *vg1-Flag*, *cyc-Flag* or *bmp7a-Flag* mRNA. TGF-beta family members are disulfide-linked dimers: under reducing conditions the disulfide bonds are broken, while under non-reducing conditions the bonds are maintained, allowing the detection of dimers. Bmp7a-Flag mature homodimers were visible under non-reducing conditions, whereas Vg1-Flag homodimers were not detected (*Figure 5B*, *Figure 5—figure supplement 1C*). Using a complementary approach, we tested whether Vg1 forms homodimers by co-immunoprecipitation. While Sqt-HA and Vg1-Flag co-precipitated, Vg1-HA and Vg1-Flag did not (*Figure 5C*). These results indicate that Vg1 does not form homodimers and might be present as monomers in the absence of Nodal.

Vg1 protein is synthesized before Nodal transcription and translation begin, raising the possibility that newly synthesized Nodal monomers bind to preexisting Vg1 monomers. Alternatively, Nodal might only heterodimerize with Vg1 protein that is co-translated with Nodal. To distinguish between these possibilities, we generated a Vg1-Dendra2 photoconvertible fusion protein and injected it at the 1-cell stage. At the 64-cell stage we photoconverted Vg1-Dendra2 from green to red, and co-injected the embryos with 5 pg of *cyc* mRNA. Imaging revealed the production of red puncta, indicating that Vg1 protein synthesized prior to Nodal synthesis was able to heterodimerize and be secreted with Nodal (*Figure 5—figure supplement 1D*). This data suggests that Nodal can heterodimerize with pre-existing Vg1.

## Vg1 and Nodal are only active when co-expressed

To determine if co-expression of Vg1 and Nodal in the same cells is required for activity, we used transplantation assays to compare target gene induction in cells co-expressing *vg1* and *cyc* versus neighboring cells expressing either *vg1* or *cyc*. Nodal target gene induction only occurred when *vg1* and *cyc* were co-expressed in the same cells (*Figure 5D* and *Figure 5—figure supplement 2A*). Analogously, deposition of *vg1* mRNA to the yolk syncytial layer (YSL) of M*vg1* mutants, where *cyc* and *sqt* are expressed endogenously, was sufficient to rescue M*vg1* mutants by morphology and gene expression (*Figure 5E* and *Figure 5—figure supplement 2B*). Confocal imaging of embryos injected with *vg1* mRNA and a fluorescent dextran into the YSL indicated that the majority of fluorescence was localized to the YSL, but a few cells in the margin also inherited the fluorescent dextran (*Figure 5—figure supplement 2C*). Thus, although *vg1* is ubiquitously expressed in the early embryo (*Helde and Grunwald, 1993*; *Peterson et al., 2013*) (*Figure 5—figure supplement 2D*), its co-localization with *cyc* and *sqt* is sufficient for its role in mesendoderm formation. Taken together, these results suggest that Vg1 and Nodal are active when expressed in the same cells, where they form heterodimers.

## Vg1 can enable rapid response to low concentrations of Nodal

The requirement for Vg1-Nodal heterodimers for mesendoderm induction raises the question of why the embryo relies on both a ubiquitous ligand, Vg1, and localized ligands, Cyc and Sqt. We developed a basic kinetic model to test the rate of Nodal homodimer formation versus Nodal-Vg1 heterodimer formation in the presence of a maternal Vg1 pool. Simulating these two conditions revealed that the preloading of inactive Vg1 monomers in the cell allows Nodal to immediately form heterodimers whereas dimer formation is delayed when Nodal must form homodimers (*Figure 6*). Thus, Vg1-Nodal heterodimers can initiate signaling more quickly than Nodal homodimers, and already at

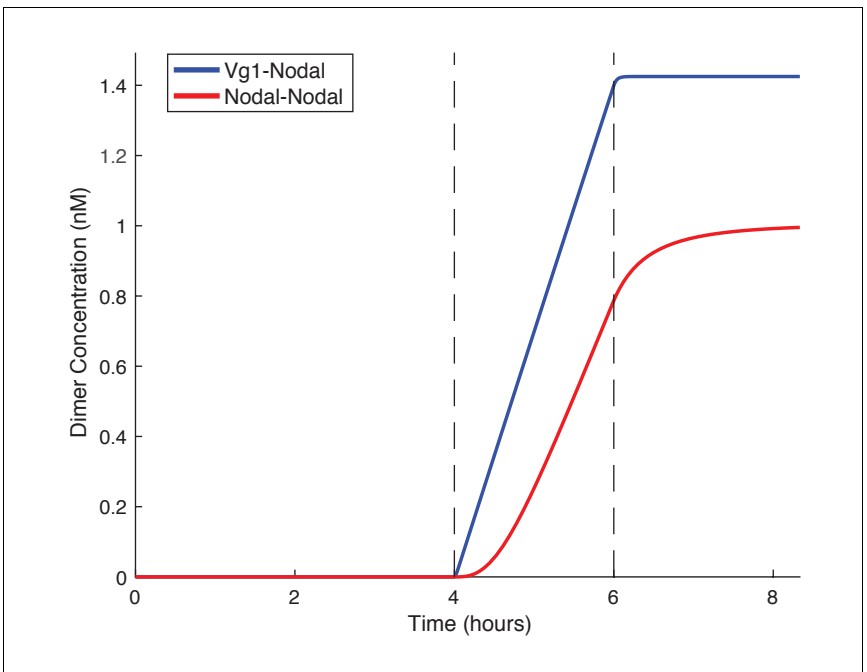

**Figure 6.** Vg1 can enable rapid response to low concentrations of Nodal. Kinetic model comparing Nodal homodimer formation in the absence of Vg1 (red line) and Vg1-Nodal heterodimer formation in the presence of a maternal Vg1 pool (blue line). For both conditions Nodal monomer production begins at 4 hpf (the onset of zygotic transcription and translation, first dotted line) and concludes after mesendodermal patterning (6 hpf, second dotted line). For the heterodimer simulation an excess of Vg1 is provided in the initial conditions.
DOI: https://doi.org/10.7554/eLife.28183.016

low Nodal levels. Even if Nodal homodimers were as active as Vg1-Nodal heterodimers, Nodal target gene induction would still be slower in the absence of maternal Vg1, because the association of two Nodal monomers is less likely at low Nodal concentrations than Vg1-Nodal dimerization (*Figure 6*). These simulations reveal that low concentrations of zygotic Nodal can be directly transformed into pathway activation via association with maternal Vg1.

## Discussion

The results in this study indicate that mesendoderm induction depends on the co-expression and heterodimer formation of Nodal and Vg1. This conclusion is based on five new findings: Vg1 is essential for mesendoderm induction; Vg1 is only processed, secreted and active in the presence of Nodal; Nodal activity, but not processing and secretion, depends on Vg1; Vg1 and Nodal form heterodimers; and Vg1-Nodal heterodimers are more active than Nodal alone. Together with previous studies, our findings suggest a unifying 5-step model for mesendoderm induction in zebrafish: (1) *vg1* mRNA is inherited from the mother and is ubiquitous in the early embryo; (2) Vg1 protein is synthesized ubiquitously and retained predominantly in the ER; (3) *cyc* and *sqt* are transcribed and translated in the YSL; (4) Cyc and Sqt form heterodimers with pre-existing Vg1, resulting in Vg1 secretion and cleavage; (5) Cyc-Vg1 and Sqt-Vg1 heterodimers activate the Nodal signaling pathway to induce mesendoderm (*Figure 7*).

### Vg1 is as essential as Nodal for mesendoderm induction

Knockdown studies in zebrafish suggested no requirement for Vg1 in mesendoderm induction (*Peterson et al., 2013*), but the loss-of-function mutants reported here reveal that Vg1 is absolutely required for the induction of head and trunk mesoderm and endoderm. Strikingly, *vg1* mutants strongly resemble Nodal signaling mutants, showing that zebrafish Vg1 has as essential a function as Nodal, which has been considered the sole mesendoderm inducer.

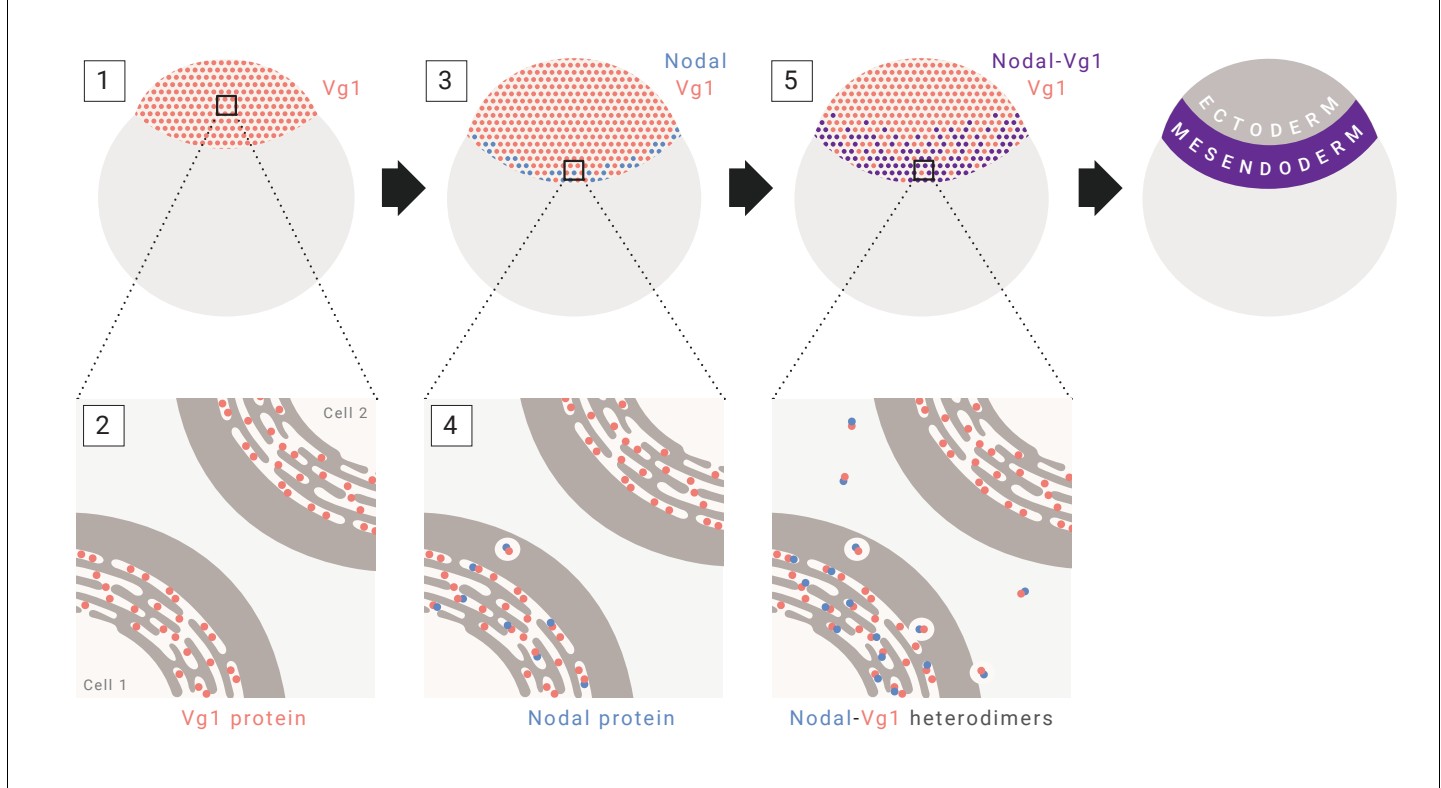

**Figure 7.** Model for mesendoderm induction in zebrafish. As described in the main text: (1) *vg1* mRNA is inherited from the mother, and is ubiquitous in the early embryo; (2) Vg1 protein is synthesized ubiquitously and retained predominantly in the ER; (3) *cyc* and *sqt* are transcribed and translated in the YSL; (4) Cyc and Sqt form heterodimers with Vg1, resulting in Vg1 secretion and cleavage; (5) Cyc-Vg1 and Sqt-Vg1 heterodimers activate the Nodal signaling pathway to induce mesendoderm.

DOI: https://doi.org/10.7554/eLife.28183.017

The results in zebrafish warrant a re-analysis of the requirements for Vg1 orthologs in other systems. For example, mouse *Gdf1;Gdf3* double mutants have incompletely penetrant mesendodermal phenotypes (*Andersson et al., 2007*). A closer comparison to *Nodal* mutants might reveal functions of GDF1 and GDF3 that are equivalent to Vg1. It is also possible that mouse Nodal is expressed at sufficiently high levels to be less dependent on GDF1/GDF3, akin to the overexpression of zebrafish *sqt*. Similarly, zebrafish *southpaw* might be expressed at high enough levels to act independently of zygotic *vg1* during left-right development. Knockdown studies in *Xenopus* have suggested that Vg1 is mainly involved in inducing notochord precursors, but not other mesendodermal progenitors (*Birsoy et al., 2006*). Mutant studies in *Xenopus* might reveal broader roles for Vg1, or alternatively additional TGF-beta-related signals such as Derrière (*Sun et al., 1999*), which has been shown to interact with Nodal (*Eimon and Harland, 2002*), might have complementary or overlapping functions with Vg1.

More generally, it is conceivable that the activities of the Nodal and Vg1/GDF1/GDF3 subfamilies are co-dependent in all contexts. This idea is not only supported by the co-dependence of Nodal and Vg1 in zebrafish mesendoderm induction reported here, but also the observation that Nodal expression coincides with the expression of Vg1 family members in numerous contexts (*Agius et al., 2000*; *Levin et al., 1995*; *Onai et al., 2010*; *Range and Lepage, 2011*; *Range et al., 2007*; *Seleiro et al., 1996*). Moreover, mouse *Gdf1* mutants display very similar left-right defects as mutants with impaired Nodal signaling (*Andersson et al., 2006*; *Cheng et al., 2003*; *Lowe et al., 2001*; *Yan et al., 1999*). It is therefore tempting to speculate that wherever and whenever Nodal subfamily members are expressed and active, they are accompanied by Vg1 subfamily members. In this scenario, Vg1/GDF1/GDF3 act in parallel with Nodal even when their expression is precedent, and any apparent upstream functions of Vg1 (*Andersson et al., 2007*; *Chen et al., 2006*;

*Rankin et al., 2000*; *Shah et al., 1997*; *Skromne and Stern, 2001*; *Tanaka et al., 2007*) are actually the result of Nodal autoinduction: Vg1 is required together with Nodal to fully activate Nodal gene expression but it is not needed for the initiation of Nodal expression, as shown in the M*vg1* mutants. Finally, Nodal and Vg1 act through the same co-receptors (*Andersson et al., 2007*; *Cheng et al., 2003*; *Fuerer et al., 2014*; *Tanaka et al., 2007*) and are both inhibited by Lefty ligands (*Agathon et al., 2001*; *Bisgrove et al., 1999*; *Chen and Shen, 2004*; *Chen and Schier, 2002*; *Cheng et al., 2004*; *Meno et al., 1999*; *1996*; *Thisse et al., 2000*; *Thisse and Thisse, 1999*). These observations suggest that Nodal signaling should henceforth be considered Nodal/Vg1 signaling.

## Vg1 is only processed, secreted and active in the presence of Nodal

Our study clarifies previously puzzling observations on the activity and processing of Vg1 that contrast with the properties of other TGF-beta-related signals: overexpression of wild-type *vg1* does not cause a phenotype (*Dale et al., 1993*; *Dohrmann et al., 1996*; *Tannahill and Melton, 1989*; *Thomsen and Melton, 1993*), neither secreted (*Dale et al., 1993*; *Tannahill and Melton, 1989*) nor processed Vg1 has been reliably detected (*Dale et al., 1989*; *Dohrmann et al., 1996*; *Tannahill and Melton, 1989*; *Thomsen and Melton, 1993*), but fusion of the Vg1 mature domain to the Activin or BMP prodomain results in processed and active Vg1 (*Dale et al., 1993*; *Dohrmann et al., 1996*; *Kessler and Melton, 1995*; *Thomsen and Melton, 1993*; *Wall et al., 2000*). Our study explains these conundrums by revealing that Vg1 is only processed, secreted and active in the presence of Nodal. Without Nodal, Vg1 is unprocessed and predominantly resides in the ER. Upon overexpression, Vg1 contributes to this inert pool, and only fusion to heterologous prodomains allows secretion, cleavage and activation in the absence of Nodal. Thus, the dependence of Vg1 processing, secretion and activity on Nodal accounts for many of the previously confusing observations. One conundrum remains: we and others have not been able to detect the processing (*Figure 3C*) or secretion (*Figure 4A*) of Vg1 at endogenous levels of Nodal. We speculate that endogenous Nodal is expressed at very low levels and in few cells, resulting in cleavage and secretion of only a small (and undetectable) fraction of the total pool of Vg1. Only upon ectopic Nodal expression, sufficiently high levels of Vg1 are processed to become detectable. The development of more sensitive detection methods is needed to directly demonstrate the cleavage, processing and secretion of endogenous Vg1.

## Vg1 and Nodal form heterodimers

Our study reveals that Nodal and Vg1 form heterodimers, and that Vg1 exists in a monomeric state prior to heterodimerization with Nodal. The initial localization of Vg1 to the ER suggests that this is the site of heterodimerization, which is consistent with previous studies of other heterodimers (*Duitman et al., 2008*; *Hurtley and Helenius, 1989*; *Jalah et al., 2013*; *Lorenz et al., 2002*; *Persson and Pettersson, 1991*; *Tu et al., 2002*). For example, in the case of the uroplakin proteins UPIb and UPIII, UPIb can autonomously exit the ER and translocate to the plasma membrane. By contrast, UPIII must heterodimerize with UPIb in the ER in order to exit and move to the plasma membrane (*Tu et al., 2002*). Although we currently favor a model in which monomeric Vg1 meets Nodal in the ER, more complex scenarios are conceivable. For example, it is unclear whether preexisting Vg1 might associate with other ER-resident proteins to maintain or prepare it in a state that allows dimerization with newly synthesized Nodal.

Our results suggest that Nodal-Vg1 heterodimers are more potent than Nodal alone: in the case of Cyc, such heterodimers seem to be required for Cyc to activate signaling, whereas Sqt-Vg1 heterodimers appear to be more active than Sqt alone (*Figures 2D, E and F*). The molecular basis of the increased activity is unknown, but based on previous studies in the BMP system, heterodimers might be necessary to assemble heteromeric combinations of two types of class I or II receptors (*Little and Mullins, 2009*).

Our results also extend and generalize the previous observation that mouse GDF1 and Nodal form heterodimers (*Fuerer et al., 2014*; *Tanaka et al., 2007*), although those studies did not address the requirement, localization, or processing of Vg1/GDF1/GDF3 during mesendoderm formation, and instead proposed that heterodimer formation might increase the potency and/or range of Nodal. Our results uncover the alternative or additional mechanism that heterodimer formation

triggers processing and secretion of Vg1 and allows Nodal to be active at physiological concentrations.

### *vg1* mRNA and protein do not need to be localized in the embryo

Our results demonstrate a novel mode to restrict TGF-beta-related protein activity through hetero-dimer formation. *vg1* mRNA and protein do not need to be localized in the embryo to restrict Vg1 activity: instead, it is the absence of Nodal that blocks Vg1 processing, secretion and activity. Indeed, zebrafish Vg1 is present ubiquitously in early embryos and *vg1* is expressed in broader domains than Nodal in all systems analyzed. The question therefore arises whether the exquisite vegetal localization of *Xenopus vg1* is important for development (*Weeks and Melton, 1987*). The localized activation of *Xenopus* Nodal genes might be sufficient to restrict mesendoderm formation to vegetal and marginal regions, but it is also possible that localized Vg1 provides an additional safe-guard to spatially restrict pathway activation. Rescue experiments similar to those reported here could address this question.

### Vg1 can enable rapid response to low concentrations of Nodal

Modeling of hetero- and homodimerization kinetics reveals that the maternal pool of Vg1 acceler-ates the onset of ligand dimerization relative to a system that relies on Nodal-Nodal dimerization alone. This could be advantageous in the embryo, where mesendoderm induction cannot initiate until after the maternal-to-zygotic transition. Although it may be counterintuitive for a spatially local-ized signal to rely on a ubiquitous signal for pathway activation, the preloading of Vg1 in the ER could be instrumental for ensuring Nodal signaling initiates in a rapid and temporally reliable man-ner. Thus, the requirement for Vg1 in the zebrafish embryo can ensure rapid and sensitive response to the low concentrations of Nodal that initiate mesendoderm induction.

### Concluding remarks

The finding that Vg1 – together with Nodal – is an endogenous mesoderm inducer resolves some of the historical controversies in the field. Vg1 was described in 1987 as a TGF-beta-related signal pres-ent at the right time and place to be a mesoderm inducer (*Weeks and Melton, 1987*), but the lack of a functional requirement raised doubts about its importance. Conversely, Activin was reported in 1990 as a TGF-beta-related signal that can induce mesoderm (*Smith et al., 1990*; *van den Eijnden-Van Raaij et al., 1990*), but its absence during early embryogenesis (*Dohrmann et al., 1993*; *Thomsen et al., 1990*), and the lack of loss-of-function phenotypes (*Hawley et al., 1995*; *Kessler and Melton, 1995*; *Matzuk et al., 1995*; *Schulte-Merker et al., 1994*; *Sun et al., 1999*), raised doubts about its importance (*Schier and Shen, 2000*). With the discovery of the essential roles of mouse Nodal (*Conlon et al., 1994*; *Zhou et al., 1993*) and zebrafish Nodal (*Feldman et al., 1998*) in mesoderm induction, the field converged to the view that Nodal is the key inducer. Our study indicates instead that Nodal-Vg1 heterodimers are the essential endogenous inducers of mes-endoderm, while Activin serves as a powerful reagent to induce mesendoderm from embryonic stem cells.

## Materials and methods

### CRISPR/Cas9-mediated mutagenesis of *vg1*

sgRNAs targeting the *vg1/dvr1/gdf3* gene were designed using CHOPCHOP (RRID:SCR_015723) (*Labun et al., 2016*; *Montague et al., 2014*) and synthesized as previously described (*Gagnon et al., 2014*) (See also *Figure 1—figure supplement 1*). *vg1* sgRNAs were co-injected with ~0.5 nL of 50 µM Cas9 protein into TLAB wild-type embryos. Injected embryos were raised to adulthood and outcrossed to TLAB adults. Clutches of embryos with potential heterozygous individ-uals were used to identify founders with germline mutations in *vg1* by extracting DNA from 10 embryos and genotyping by MiSeq sequencing. The offspring of confirmed founders were raised to adulthood and genotyped to identify heterozygous *vg1* adults. Heterozygous *vg1* mutants were intercrossed to generate zygotic homozygous (Z*vg1*) fish. For maintaining the *vg1* mutant line, homozygous Z*vg1* male fish were crossed to heterozygous female fish, and the resulting progeny

were genotyped to identify Z*vg1* adults. To generate maternal *vg1* (M*vg1*) mutants, TLAB wild-type male fish were crossed to homozygous Z*vg1* female fish.

## Genotyping of *vg1* mutants

Two deletion alleles of 8 bp and 29 bp (*vg1*[a164] and *vg1*[a165] respectively) were recovered in the first exon of *vg1* from the sgRNA targeting the sequence GGGTCAGAAGACAGGCTCTGAGG. Genomic DNA was extracted using the HotSHOT method (*Meeker et al., 2007*) and PCR was performed using standard conditions (see primer sequences below), followed by Sanger sequencing or MiSeq sequencing for the 8 bp allele (*Gagnon et al., 2014*) or 2% gel electrophoresis for the 29 bp allele.

## Cloning of expression constructs

The *vg1* CDS sequence was PCR amplified from a high-stage cDNA library and cloned into the pSC vector (Agilent) with a beta-globin 5'UTR and an SV40 3'UTR using Gibson assembly (*Gibson et al., 2009*) to generate pSC-*vg1*. To generate pCS2(+)-*cyc* and pCS2(+)-*sqt*, the *cyc* and *sqt* CDS sequences were PCR amplified from a high-stage cDNA library and cloned into the pCS2(+) vector using Gibson assembly. To generate non-cleavable forms of *vg1* and *vg1-sfGFP*, site-directed mutagenesis was used to replace the RSRRKR cleavage site with SQNTSN using a Q5 Site-Directed Mutagenesis Kit (NEB).

## Cloning of fusion and epitope tag constructs

All superfolder GFP (sfGFP) (*Pédelacq et al., 2006*), RFP, Dendra2 and pHluorin2 (*Mahon, 2011*) fusion constructs were generated by PCR-based methods and cloned into the pCS2(+) vector using Gibson assembly. Flag (DYKDDDDK) and HA tag (YPYDVPDYA) sequences were inserted by site-directed mutagenesis of pSC-*vg1*, pCS2(+)-*cyc* and pCS2(+)-*sqt*. For Vg1 fusions, sequences encoding the fluorescent protein or Flag tag were inserted downstream of the cleavage site (RSRRKR) with a GSTGTT linker separating the prodomain and fluorescent protein, and a GS linker separating the fluorescent protein and the Vg1 mature domain. For Cyc fusions, sequences encoding the fluorescent proteins or HA tag were inserted two amino acids downstream of the cleavage site (RRGRR) (*Müller et al., 2012*). For Sqt fusions, fluorescent protein and HA tag sequences were inserted 10 amino acids downstream of the cleavage site (RRHRR) with a GSTGTT linker separating the prodomain and fluorescent protein, and a GS linker separating the fluorescent protein and the mature domain (*Müller et al., 2012*).

## mRNA synthesis and microinjection

Vectors were linearized by digestion with NotI (pCS2(+) vectors) or XhoI (pSC vectors). Capped mRNAs were synthesized using the SP6 or T7 mMessage Machine Kits (ThermoFisher), respectively. For *in situ* hybridization, immunoblot, imaging and qPCR experiments, embryos were dechorionated using 1 mg/ml Pronase (Protease type XIV from Streptomyces griseus, Sigma) prior to injection, and subsequently cultured in agarose-coated dishes. Embryos were injected at the 1-cell stage unless otherwise stated.

## Zebrafish husbandry

Zebrafish embryos were grown at 28°C and staged according to (*Kimmel et al., 1995*). Embryos were cultured in blue water (250 mg/L Instant Ocean salt, 1 mg/L methylene blue in reverse osmosis water adjusted to pH 7 with NaHCO$_3$).

## Morphological analysis of *vg1* mutant phenotypes

Embryos were analyzed for mutant phenotypes at 28–32 hpf. For imaging, embryos were anesthetized in Tricaine (Sigma) and mounted in 2% methylcellulose then imaged using a Zeiss SteREO Discovery.V12 microscope.

## Live imaging

Embryos were raised to sphere stage and mounted in 1% low gelling temperature agarose (Sigma) on glass-bottomed dishes (MatTek) with the animal pole facing the glass. Imaging was performed on Zeiss LSM 700 and LSM 880 inverted confocal microscopes.

## Photoconversion

Embryos were injected at the 1-cell stage with 100 pg of *vg1-Dendra2* mRNA then grown at 28°C to the 64-cell stage and injected with 5 pg of *cyc* mRNA into six locations in the embryo. Embryos were mounted in 1% low gelling temperature agarose and photoconverted with 2 min of UV light at 10x magnification on the Zeiss LSM 700 inverted confocal microscope. The embryos were incubated at 28°C for 30 min before imaging on the LSM 700 microscope over a period of 2 hours.

## Image adjustments

Images were processed in FIJI/ImageJ (*Schindelin et al., 2012*). Brightness, contrast and color balance was applied uniformly to images.

## *in situ* hybridization and DAB staining

Embryos were fixed in 4% formaldehyde overnight at room temperature (50% epiboly or younger) or at 4°C (embryos older than 50% epiboly). Whole mount *in situ* hybridizations were performed according to standard protocols (*Thisse and Thisse, 2008*). DIG-labeled antisense RNA probes against *cmlc2, spaw, gsc, lft1, ntl, sox32, cyc, sqt* and *vg1* were synthesized using a DIG Probe Synthesis Kit (Roche). NBT/BCIP/Alkaline phosphatase-stained embryos were dehydrated in methanol and imaged in benzyl benzoate:benzyl alcohol (BBBA) using a Zeiss Axio Imager.Z1 microscope. For DAB staining, embryos were rehydrated in PBST after completing the *in situ* protocol, and blocked in 10% normal goat serum/1% DMSO before incubation in primary antibody overnight (1:400 rabbit anti-GFP-HRP, ThermoFisher A10260, RRID:AB_2534022). Embryos were washed multiple times in PBST, incubated in DAB solution (KPL #71-00-48), and dehydrated before imaging in BBBA.

## Transplantation and YSL injection

For transplantation experiments, donor embryos were injected with 50 pg of *cyc* mRNA and/or *vg1* mRNA and 50 pg of *GFP* mRNA and grown to sphere stage (4 hpf). At sphere stage, cells were transplanted from donor embryos to host embryos, and host embryos were grown to shield stage before fixation for *in situ* hybridization. For YSL injections, 1000-cell stage embryos were injected through the chorion into the YSL with approximately 100 pg of *vg1* mRNA and 500 pg of Alexa Fluor 488 dextran (ThermoFisher).

## Immunoblotting

Embryos were injected at the 1-cell stage with 50 pg of each mRNA (unless otherwise stated) and grown to early gastrulation (50% epiboly). 8 embryos per sample were manually deyolked with forceps and frozen in liquid nitrogen. The samples were boiled at 95°C for 5 min with 2x SDS loading buffer (10 μL) and DTT (reducing gels only, 150 mM final concentration) and then loaded onto Any kD protein gels (Bio-Rad). Samples were transferred to polyvinylidene fluoride (PVDF) membranes (GE Healthcare). Membranes were blocked in 5% non-fat milk (Bio-Rad) in TBST and incubated overnight at 4°C in primary antibodies (1:5000 rabbit anti-GFP, ThermoFisher A11122, RRID:AB_221569; 1:2000 rabbit anti-Flag, Sigma F7425, RRID:AB_439687). Proteins were detected using HRP-coupled secondary antibody (1:15,000 goat anti-rabbit, Jackson ImmunoResearch Labs 111-035-144, RRID: AB_2307391). Chemiluminescence was detected using Amersham ECL reagent (GE Healthcare).

## Co-immunoprecipitation

Dechorionated embryos were injected at the 1-cell stage with 5, 20 or 50 pg of mRNA encoding epitope-tagged constructs and grown to 50% epiboly. 50–100 embryos were transferred to 400 μL of cold lysis buffer (50 mM Tris at pH 7.5, 150 mM NaCl, 1 mM EDTA, 10% glycerol, 1% Triton X-100 and protease inhibitors, Sigma 11836170001) and crushed using a homogenizer and disposable pestles before incubation on ice for 30 min with vortexing every 5 min. Samples were spun at maximum speed at 4°C for 30 min and the supernatant was transferred to tubes containing 50 μL of anti-HA affinity matrix (Roche 11815016001, RRID:AB_390914) that was pre-washed twice in lysis buffer. Samples were placed on a rotating platform at 4°C overnight. The matrix was spun down for 2 min at 3000 rcf and washed in 600 μL of cold wash buffer (50 mM Tris at pH 7.5, 150 mM NaCl, 1% Triton X-100 and protease inhibitors) 5 times. 2x SDS loading buffer and DTT (150 mM final concentration) was added to the matrix in 10 μL of wash buffer. Immunoblots were performed as above.

## qPCR

Embryos were injected at the 1-cell stage with *sqt*, *cyc* and/or *vg1* mRNAs and grown to 50% epiboly. For the *sqt* experiment (**Figure 2E**) 2 sets of 10 embryos were collected per condition; for the *cyc* experiment (**Figure 2F**) 2 sets of 12 embryos were collected per condition. Embryos were flash frozen in liquid nitrogen and RNA was extracted using an E.Z.N.A. Total RNA Kit (Omega) and reverse transcription was carried out using an iScript cDNA Synthesis Kit (Bio-Rad). qPCR reactions were run on a CFX96 machine (Bio-Rad) using iTaq Universal SYBR Green Supermix (Bio-Rad) and 0.25 µM of primers (see primer sequences below). Gene expression levels were calculated relative to a reference gene, *ef1a*. The mean and standard error of the mean was plotted for each condition. Two technical replicates in addition to biological replicates were used per condition. Both experiments were performed multiple times.

## Primer sequences

| | |
|---|---|
| vg1_genotype_F | CCTGTGTGTGTTCTTTGCTCTG |
| vg1_genotype_R | CTGTTTAAAGATTTTCCACATCTGTG |
| ef1a_qPCR_F | AGAAGGAAGCCGCTGAGATGG |
| ef1a_qPCR_R | TCCGTTCTTGGAGATACCAGCC |
| lft1_qPCR_F | GAGATGGCCAAGTGTGTCCA |
| lft1_qPCR_R | CTGCAGCACATTTCACGGTC |

## Modeling

In the 'primed model', Vg1 is already present in excess when Nodal production begins. This model describes the dynamics of Nodal monomers (*N*), Vg1 monomers (*V*) and Nodal-Vg1 dimers (*D*).

$$\frac{dN}{dt} = \lambda_N - \beta_N N - \lambda_D NV$$

$$\frac{dV}{dt} = -\beta_V V - \lambda_D NV$$

$$\frac{dD}{dt} = \lambda_D NV$$

Assumptions: constitutive production of *N* (rate $\lambda_N$), first-order degradation (component half-lives of $ln2/\beta_N$ and $ln2/\beta_V$, respectively) and bimolecular heterodimerization with rate $\lambda_D NV$. Vg1 is assumed to be maternally deposited, and is thus provided to the system via the initial conditions.

In the 'cold-start' model, Nodal monomers accumulate and dimerize after the onset of Nodal production. This model describes the dynamics of Nodal monomers (*N*), Vg1 monomers (*V*) and Nodal-Nodal dimers (*D*).

$$\frac{dN}{dt} = \lambda_N - \beta_N N - \lambda_D N^2$$

$$\frac{dD}{dt} = \lambda_D N^2$$

Assumptions: constitutive production of *N* (rate $\lambda_N$), first-order degradation (component half-life of $ln2/\beta_N$) and bimolecular homodimerization with rate $\lambda_D N^2$.

### Simulation

The system begins with an excess of *V*, representing maternally deposited Vg1. All other component concentrations begin at 0, and Nodal monomer production is assumed to be off. Nodal monomer production begins at 4 hpf (the onset of zygotic transcription and translation) and concludes after

mesendodermal patterning (6 hpf). The concentration of dimer (either Vg1-Nodal heterodimer or Nodal-Nodal homodimer) was plotted.

| Parameter | Description | Value | Units |
|---|---|---|---|
| $\lambda_N$ | Nodal synthesis rate | 0, t < 4 hr, t > 6 hr<br>$2 \times 10^{-13}$, 4 hr < t < 6 hr<br>(*Müller et al., 2012*) | M s$^{-1}$ |
| $\beta_N$ | Nodal degradation rate | $1.16 \times 10^{-4}$ (*Müller et al., 2012*) | s$^{-1}$ |
| $\beta_V$ | Vg1 degradation rate | $1.16 \times 10^{-4}$ (*Müller et al., 2012*) | s$^{-1}$ |
| $\lambda_D$ | Dimerization rate | $1 \times 10^6$ (*Gerstle and Fried, 1993*;<br>*Kohler and Schepartz, 2001*;<br>*Northrup and Erickson, 1992*) | M$^{-1}$ s$^{-1}$ |
| V | Vg1 initial concentration | 100 | nM |

# Acknowledgements

The authors would like to thank Nate Lord, Jeff Farrell and Constance Richter for helpful comments on the manuscript, Nate Lord for modeling, Toby Montague for graphics advice, Fred Rubino for immunoblot advice and Andrea Pauli for mentorship. We thank Rebecca Burdine and Joe Yost for collegial interactions and for coordinating publications. The BMP7a-Flag construct was a gift of Mary Mullins. This project was supported by NIH GM056211 (AFS) and a National Defense Science and Engineering Graduate (NDSEG) Fellowship (TGM).

# Additional information

## Funding

| Funder | Grant reference number | Author |
|---|---|---|
| National Institutes of Health | GM056211 | Alexander F Schier |
| American Society for Engineering Education | National Defense Science and Engineering Graduate Fellowship | Tessa Grace Montague |

The funders had no role in study design, data collection and interpretation, or the decision to submit the work for publication.

## Author contributions

Tessa G Montague, Conceptualization, Formal analysis, Validation, Investigation, Visualization, Methodology, Writing—original draft, Writing—review and editing; Alexander F Schier, Conceptualization, Formal analysis, Supervision, Funding acquisition, Writing—review and editing

## Author ORCIDs

Tessa G Montague http://orcid.org/0000-0002-5918-6327
Alexander F Schier http://orcid.org/0000-0001-7645-5325

## Ethics

Animal experimentation: All vertebrate animal work was performed at the facilities of Harvard University, Faculty of Arts & Sciences (HU/FAS). The HU/FAS animal care and use program maintains full AAALAC accreditation, is assured with OLAW (A3593-01), and is currently registered with the USDA. This study was approved by the Harvard University/Faculty of Arts & Sciences Standing Committee on the Use of Animals in Research & Teaching under Protocol No. 25-08.

## Decision letter and Author response

Decision letter https://doi.org/10.7554/eLife.28183.023

Author response https://doi.org/10.7554/eLife.28183.024

## Additional files

**Supplementary files**

• Source code 1. Nodal homodimerization versus Vg1-Nodal heterodimerization simulation.
DOI: https://doi.org/10.7554/eLife.28183.018

• Source code 2. Ordinary differential equations for Nodal homodimer simulation.
DOI: https://doi.org/10.7554/eLife.28183.019

• Source code 3. Ordinary differential equations for Vg1-Nodal heterodimer simulation.
DOI: https://doi.org/10.7554/eLife.28183.020

• Transparent reporting form
DOI: https://doi.org/10.7554/eLife.28183.021

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
