## [Decision Letter]

Thank you for submitting your article "Vg1-Nodal heterodimers are the endogenous inducers of mesendoderm" for consideration by *eLife*. Your article has been reviewed by three peer reviewers, and the evaluation has been overseen by a Reviewing Editor and Didier Stainier as the Senior Editor. The following individuals involved in review of your submission have agreed to reveal their identity: Lilianna Solnica-Krezel (Reviewer #3).

The reviewers have discussed the reviews with one another and the Reviewing Editor has drafted this decision to help you prepare a revised submission.

1) In revising the manuscript, further experiments are necessary to clarify some basic issues about the model. For example, why Vg1 remains as monomers and thus available to dimerize with Nodal. The suggested experiment is to perform additional WB under-non reducing conditions to test this. It would also be advisable to experimentally clarify where exactly in the cell heterodimerization is occurring. Additionally two of the reviewers raise the issue as to why heterodimers are required to mediate nodal signaling versus homodimers? It would be important to show a direct interaction between these heterodimers and the nodal receptor complexes. An alternative approach would be to over-express different relative amounts of Vg1 and Nodal in Vg1 mutants and use Phospho-Smad2 activity as a read-out (as suggested by reviewer 2).

2) Two of the reviewers raise concerns with the Vg1 mutant rescue by YSL injection of Vg1. In particular reviewer 3 requests that the experiment be repeated using lineage tracing to show that the injection is confined to the YSL, ideally with the inclusion of Smad2 staining as further verification.

Typically, when requesting revisions, *eLife* decision letters only include a summary of the major concerns discussed by the editors and the referees, but we considered that the authors would benefit from receiving the full reviews verbatim in this occasion so that they can attend to as many of the points raised as possible, and also see that the reviewers were uniformly very positive.

Reviewer #1:

This is a very important paper to the field showing that Vg1-Nodal heterodimers are essential to induce mesendoderm in zebrafish and likely more broadly in vertebrates. Vg1 has a long history in the field, but its endogenous function has been previously untested in a rigorous manner. This paper by Montague and Schier shows convincingly that Vg1 can only be processed and secreted when it heterodimerizes with Nodal, and that Nodal homodimers do not function in the absence of Vg1, unless highly overexpressed (Sqt only). Strong evidence has previously implicated Gdf-Nodal heterodimers in left-right patterning, so it is likely that Nodal-Vg1 heterodimers function exclusively in signaling. The results are all convincing but some points need addressing to improve the manuscript.

The kinetic model assumes that Vg1 would remain as monomers ready to heterodimerize with Nodal, which is a major assumption of the model and should be mentioned. Vg1 alone would be expected to homodimerize if Nodal is not present. Do the authors have evidence for Vg1 remaining as a monomer in the absence of Nodal? Non-reducing Western blot conditions could provide evidence for this.

The authors should also provide more information on where dimerization is known to occur (Golgi?) and how it is regulated to provide more plausibility to the model? The authors have stated that unprocessed Vg1 resides in the ER but haven't provided evidence to support this. They should either provide data to support this ER retention or soften their statements throughout.

The starting level of Vg1 in the model at 4 hpf is assumed to be 1000 nM, which is extremely high, especially considering that the mRNA is at very low levels at this point, although protein could be higher but are unknown. Provide modeling results when Vg1 monomers are at 500 nm or a more reasonable level of 50 or 100 nm.

It would be helpful if the authors could comment on why Vg1-Nodal heterodimers are required for signaling. Readers will want to know possible functions for a heterodimer that the Nodal homodimer cannot provide or at least comment that it is unclear at this point. Perhaps I missed this in the paper.

It would be helpful if the authors commented further on the rescue of Mvg1 mutants by YSL injection of *vg1* mRNA at the 1000-cell stage, a stage after zygotic transcription has initiated. This suggests that a large maternal pool of unprocessed Vg1 is not essential for Nodal-Vg1 heterodimer formation and function in mesendoderm induction. Additionally comment on the sufficiency of Nodal-Vg1 production in the YSL, which may have been previously performed with Nodal. Is the zygotic marginal domain of *sqt/cyc* RNA expression not essential then in these conditions?

Additionally, Vg1 is localized during oogenesis both in *Xenopus* (vegetal pole) and at animal pole in zebrafish. Since the vegetal localization in *Xenopus* is mentioned and discussed multiple times, the animal localization of *vg1* RNA during oogenesis is worth noting to readers as well. Typically one reason these RNAs are localized during oogenesis is to translationally repress them, which is also worth mentioning, since the authors discuss reasons for localization in the Discussion.

Reviewer #2:

Montague and Schier describe the role of Vg1 in mediating Nodal signaling during zebrafish gastrulation. They show that a Vg1 mutant phenocopies Nodal signaling defective mutants, such as the *oep* and the *cyc/squint* double mutants, pointing to the possibility that Vg1 is necessary for Nodal signaling during gastrulation. They also show that Vg1 is maternally provided in an inactive form that resides intracellularly and becomes active only when Nodal ligands are produced. They then show that Nodal ligands bind Vg1 and propose a mechanism by which Nodal/Vg1 heterodimers mediate effective nodal signaling.

The study is of wide interest as it provides new insight on the Nodal signaling pathway, a conserved pathway involved in embryonic patterning and organogenesis. The experimental data are solid and clearly presented, however a few points need to be clarified prior to publication.

1) The authors claim that Nodal/Vg1 heterodimers mediate Nodal signaling. This statement requires at least direct prove that such heterodimers bind to nodal receptors and, ideally, that they mediate receptor activation. This could be shown by IP experiments showing binding of both Cyc and Vg1 to phosphorylated Nodal receptors.

2) The authors show that Vg1 binds to Cyc and Sqt in vivo by colocalization of tagged versions of these ligands in the extracellular space (Figure 4). These colocalizations should be quantified to show what fraction of extracellular Vg1 is bound to Nodal ligands and vice-versa.

Also, in Figure 4) Co-expression of cyc and Vg1-GFP changes Vg1 localization from completely intracellular to completely extracellular. Same for co-expression with Sqt (Figure 4). Is this true? Or is it due to differences in Image exposure? A quantification of this should be provided. Also it would be interesting to know if the fraction of extracellular Vg1 depends on the amount of Nodal ligand provided.

b) The distribution of Sqt-dsRED in Figure 4 looks very different from that of Sqt-GFP in Figure 2 and from that of Vg-1 in the presence of Sqt in Figure 2. Why is this so?

3) If Nodal signaling is mediated by heterodimers of Nodal and Vg1, the extent of Nodal signaling should depend on their stoichiometry. It would be interesting to overexpress different relative amounts of Vg1 and Nodal ligand in Vg1 mutant and analyse Nodal signaling activation (via Smad2 nuclear localization for instance).

4) The authors provide a simple theoretical model to show how the presence of maternally contributed Vg1 speeds up Nodal signaling upon production of Nodal ligands. I feel this is not particularly revealing: the idea that one component of a pathway being readily available speeds up signaling does not require modeling to be put forward. Moreover, the model is based on parameters measured or inferred for the Nodal ligands, which do not necessarily hold true for Vg1.

Reviewer #3:

Montague and Schier report on their studies of the developmental roles of the zebrafish Vg1 homolog, a secreted TGFb ligand, which has been proposed to play key roles in specification of mesendoderm in frogs and mouse. However, the specific roles and underlying molecular mechanisms remain disputable. Taking a rigorous reverse genetic approach, and generating two different frame-shifting indel *vg1* zebrafish mutants, the authors demonstrate an essential role of maternal *vg1* in mesendoderm induction: whereas zygotic *vg1* mutants appear normal (but what about left-right asymmetry?), maternal mutant embryos phenocopy defects observed in the famous Nodal signaling deficient scenarios. The authors provide genetic, biochemical and cell biological evidence that Vg1 processing requires Nodal, and that Vg1 and Nodal (Cyc or Sqt) form heterodimers. Modeling is used to argue that the ubiquitously expressed *vg1* allows for faster Nodal signaling. The manuscript clarifies a large body of often incongruent data and models from various vertebrate model systems and advances our understanding of both Vg1 and Nodal signaling during development. Therefore, the manuscript should be of interest to the broad developmental biology community. The manuscript is in general logically presented but scarcity of experimental detail make it difficult to evaluate the presented results. Several conclusions need to be strengthened by additional experiments, and some experiments and their interpretation need clarification.

1) Do the indel *vg1* mutations lead to reduction of maternal *vg1* message?

2 Are zygotic *vg1* mutant truly aphenotypic? What about LR symmetry?

3) Based on the ability of 50 pg of RNA encoding Vg1-NC not to rescue Mvg1 mutant phenotype the authors conclude that endogenous cleavage is required for Vg1 function appears too strong. Higher RNA doses should be tested.

4) Immunoblot experiments presented in Figure 3 should be quantified; there are no loading controls in these experiments.

5) Experiments testing the intracellular localization of Vg1(Cyc;Sqt)-sfGFP would be more compelling if RNAs encoding these molecules would be injected with RNA encoding membraneRFP.

6) What RNA doses were injected in embryos shown in Figure 4?

7) Experiments presented in Figure 4, would be more informative if *sqt (bmp*) RNAs were co-injected with a lineage tracer. There are two cells in the *sqt* panel showing intracellular Vg1-sfGFP, likely these cells had little/no Sqt, what would further strengthen the authors' conclusions.

8) "…we performed co-immunoprecipitation experiments and found that Vg1 co-immunoprecipitated with Cyc or Sqt". This sentence could be understood that endogenous proteins were analyzed. Key experimental information is missing about how the relevant tagged proteins were expressed, at what stage of development co-IPs were performed. The co-IP with Cyc-HA is not convincing.

9) The conclusion that co-expression of Vg1 and Nodal in the same cells is required for activity, based on the experiments presented in Figure 5 is intriguing, especially for injection of *vg1* RNA into the YSL of Mvg1 mutants. Whereas it has been previously shown that Nodal expression in YSL is sufficient to induce mesoderm and endoderm in the overlying blastoderm, this experiment would indicate that Cyc-Vg1 and Sqt-Vg1 heterodimers secreted from YSL are sufficient to create Nodal activity gradient to not only induce but also pattern the blastoderm. However, this experiment raises many questions: has injection been truly confined to the YSL (there is no lineage tracing data presented and a sceptic could argue that the 13 embryos that were rescued express the ligands also in the blastoderm due to ineffective injection? The evidence that expression is confined to the YSL is essential. It would be also important to show that Smad2 activity in such embryos parallels WT pattern shown by the authors in their earlier publications. It would be also important to show that the ligands targeted by RNA injection to the YSL can be seen in extracellular spaces in the blastoderm.

10) How does the above result fit with the models of Nodal autoinduction and self-inhibition? In the model presented in Figure 7 and in the Discussion, the authors see an important role of Sqt and Cyc expressed at the embryonic margin not just in the YSL.

---

## [Author Response]

1) In revising the manuscript, further experiments are necessary to clarify some basic issues about the model. For example, why Vg1 remains as monomers and thus available to dimerize with Nodal. The suggested experiment is to perform additional WB under-non reducing conditions to test this.

We thank the reviewers for asking us to clarify the monomeric state of Vg1. We present two forms of evidence that suggest Vg1 remains as monomers prior to heterodimerization with Nodal. First, non-reducing western blots with Vg1-Flag (and Bmp7a-Flag as a positive control) show that while Bmp7a forms homodimers under non-reducing conditions, Vg1 does not. Second, by co-IP we detect an interaction between Sqt-HA and Vg1-Flag, but not between Vg1-HA and Vg1-Flag, indicating that Vg1 does not homodimerize. In addition, a previous study found that Vg1 does not form homodimers, and predominantly exists in a monomeric state (Dale et al. EMBO J 1993). We have now referred to this study in the text.

It would also be advisable to experimentally clarify where exactly in the cell heterodimerization is occurring.

We agree with the reviewers that it would be interesting to visualize where in the cell heterodimerization is occurring. Unfortunately, it is currently not feasible to perform dynamic imaging of the two proteins at the high spatial resolution required to detect heterodimer formation. In the revised manuscript, we raise this question for future studies and discuss previous studies that indicate that the endoplasmic reticulum is a strong candidate for the site of heterodimerization.

Additionally two of the reviewers raise the issue as to why heterodimers are required to mediate nodal signaling versus homodimers?

We thank the reviewers for raising this point. We have clarified in the text that heterodimers are most likely required to mediate Nodal signaling because they are more potent than homodimers. We have provided additional experimental evidence to support this (Figure 2 and our last response to point 1 below).

It would be important to show a direct interaction between these heterodimers and the nodal receptor complexes. An alternative approach would be to over-express different relative amounts of Vg1 and Nodal in Vg1 mutants and use Phospho-Smad2 activity as a read-out (as suggested by reviewer 2).

We thank the reviewers for these suggestions. We have overexpressed *cyc* with different levels of *vg1* and performed qPCR to show that the extent of Nodal signaling (measured by *lft1* expression) increases with higher levels of co-expression of *vg1*. We found phospho-Smad2 to be insufficiently quantitative for this experiment. We agree that showing a direct interaction between the heterodimers and Nodal receptor complexes will be an important experiment. However, testing receptor-ligand interactions is currently technically very challenging because the full Nodal receptor complement has not yet been identified and, with the exception of the co-receptor Oep, the zebrafish Nodal receptors have not been defined genetically. We will try to address this point in a follow up study.

2) Two of the reviewers raise concerns with the Vg1 mutant rescue by YSL injection of Vg1. In particular reviewer 3 requests that the experiment be repeated using lineage tracing to show that the injection is confined to the YSL, ideally with the inclusion of Smad2 staining as further verification.

We thank the reviewers for drawing our attention to this point. In response to the reviewers’ suggestions we performed confocal imaging on embryos co-injected with *vg1* and a fluorescent dextran into the YSL at the 1000-cell stage. While the vast majority of fluorescence was localized to the YSL, we found that occasional cells at the margin contain some fluorescent signal. We have therefore revised the text to state that the mRNA was present primarily but not exclusively in the YSL. We have further verified the rescue of the M*vg1* phenotype with gene expression markers of endoderm and mesoderm. We find that while some embryos achieve almost complete rescue, the rescue is not fully penetrant. These results suggest that maternal *vg1* is more potent when available earlier and/or more broadly.

Typically, when requesting revisions, eLife decision letters only include a summary of the major concerns discussed by the editors and the referees, but we considered that the authors would benefit from receiving the full reviews verbatim in this occasion so that they can attend to as many of the points raised as possible, and also see that the reviewers were uniformly very positive.

Thank you for sharing these comments. They helped us further improve the study.

Reviewer #1:[…] The kinetic model assumes that Vg1 would remain as monomers ready to heterodimerize with Nodal, which is a major assumption of the model and should be mentioned. Vg1 alone would be expected to homodimerize if Nodal is not present. Do the authors have evidence for Vg1 remaining as a monomer in the absence of Nodal? Non-reducing Western blot conditions could provide evidence for this.

We thank the reviewer for raising this important issue. As detailed in our first response to point 1 above, we now provide evidence that Vg1 does not form homodimers.

One potential criticism of this model is that Nodal might preferentially heterodimerize with Vg1 protein that is co-translated with Nodal rather than with Vg1 that has already been translated and is latent in the ER. To test this model we injected *vg1-Dendra2* mRNA at the 1-cell stage, and photoconverted the Vg1-Dendra2 protein at the 64-cell stage shortly after injecting *cyc* mRNA. This allowed us to distinguish the early Vg1 protein (red) from the protein co-translated with Cyc (green). Live imaging of these embryos revealed the presence of red puncta, indicating that the Vg1 protein translated prior to Nodal expression can heterodimerize with Nodal (see Figure 5—figure supplement 1).

The authors should also provide more information on where dimerization is known to occur (Golgi?) and how it is regulated to provide more plausibility to the model? The authors have stated that unprocessed Vg1 resides in the ER but haven't provided evidence to support this. They should either provide data to support this ER retention or soften their statements throughout.

We thank the reviewer for this comment. We now provide published examples of where in the cell proteins heterodimerize, and find this to predominantly occur in the ER (Hurtley and Helenius, Annu. Rev. Cell Biol.1989). For example, in the case of uroplakins, similarly to Vg1 and Nodal, 1 of the 4 uroplakin proteins is able to independently exit the ER, while the others can only exit upon heterodimerization (Tu et al. Mol. Biol. Cell 2002).

The localization pattern of Vg1 is highly indicative of ER retention (see references in the text and Tu et al. 2002). To further clarify this point we expressed an ER protein from yeast (Sec61beta-GFP) in zebrafish embryos. Unfortunately the protein only weakly expressed in zebrafish, and therefore did not provide the resolution required to clearly indicate its subcellular localization. To avoid overstating the data we have adjusted the text to state that Vg1 is “predominantly” localized in the ER.

The starting level of Vg1 in the model at 4 hpf is assumed to be 1000 nM, which is extremely high, especially considering that the mRNA is at very low levels at this point, although protein could be higher but are unknown. Provide modeling results when Vg1 monomers are at 500 nm or a more reasonable level of 50 or 100 nm.

We thank the reviewer for drawing our attention to this. We have adjusted the starting concentration of *vg1* in the model to be 100 nM and obtained similar results as before.

It would be helpful if the authors could comment on why Vg1-Nodal heterodimers are required for signaling. Readers will want to know possible functions for a heterodimer that the Nodal homodimer cannot provide or at least comment that it is unclear at this point. Perhaps I missed this in the paper.

We agree with the reviewer that we should make this point more clearly. Our data indicates that heterodimers are more potent than homodimers (see Figure 2); therefore the simple explanation is that the embryo uses heterodimers because they are more active. We have clarified this in the text and refer to the studies by Mullins and colleagues who show that Bmp2-Bmp7 heterodimers are more potent than a combination of Bmp2 homodimers and Bmp7 homodimers (Little and Mullins, Nat. Cell Biol. 2009).

An additional explanation is that maternal *vg1* allows embryos to signal at lower concentrations of Nodalbecause the embryo already has Vg1 protein when *nodal* is transcribed and translated. This point is addressed in Figure 6 with a kinetic model of homodimer versus heterodimer formation.

It would be helpful if the authors commented further on the rescue of Mvg1 mutants by YSL injection of vg1 mRNA at the 1000-cell stage, a stage after zygotic transcription has initiated. This suggests that a large maternal pool of unprocessed Vg1 is not essential for Nodal-Vg1 heterodimer formation and function in mesendoderm induction. Additionally comment on the sufficiency of Nodal-Vg1 production in the YSL, which may have been previously performed with Nodal. Is the zygotic marginal domain of sqt/cyc RNA expression not essential then in these conditions?

We thank the reviewer for this suggestion. Please see point 2 above.

Additionally, Vg1 is localized during oogenesis both in Xenopus (vegetal pole) and at animal pole in zebrafish. Since the vegetal localization in Xenopus is mentioned and discussed multiple times, the animal localization of vg1 RNA during oogenesis is worth noting to readers as well. Typically one reason these RNAs are localized during oogenesis is to translationally repress them, which is also worth mentioning, since the authors discuss reasons for localization in the Discussion.

We thank the reviewer for pointing this out. We have added this to the text.

Reviewer #2:[…] 1) The authors claim that Nodal/Vg1 heterodimers mediate Nodal signaling. This statement requires at least direct prove that such heterodimers bind to nodal receptors and, ideally, that they mediate receptor activation. This could be shown by IP experiments showing binding of both Cyc and Vg1 to phosphorylated Nodal receptors.

We thank the reviewer for this suggestion. Please see our last response to point 1 above.

2) The authors show that Vg1 binds to Cyc and Sqt in vivo by colocalization of tagged versions of these ligands in the extracellular space (Figure 4). These colocalizations should be quantified to show what fraction of extracellular Vg1 is bound to Nodal ligands and vice-versa.

We thank the reviewer for this suggestion. We have included z-stacks of Vg1-sfGFP and Cyc-RFP co-expression to provide a better indication of the extent of co-localization of the two proteins (see Figure 4—figure supplement 1). Unfortunately, a meaningful quantification of co-localization would be difficult to interpret in this case since Vg1 and Cyc are tagged with two different proteins that have different fluorescent properties.

Also, in Figure 4) Co-expression of cyc and Vg1-GFP changes Vg1 localization from completely intracellular to completely extracellular. Same for co-expression with Sqt (Figure 4). Is this true? Or is it due to differences in Image exposure?

We thank the reviewer for this observation. We adjusted the laser power when capturing these images to avoid saturation but have provided additional images captured with the same laser power to give a better idea of the relative brightness of the intracellular versus extracellular fractions of Vg1 in the presence of the two Nodal ligands (Figure 4—figure supplement 1). We find there is variation among embryos in the degree of secretion, but usually when the embryo displays puncta, the majority of Vg1 becomes extracellular.

A quantification of this should be provided. Also it would be interesting to know if the fraction of extracellular Vg1 depends on the amount of Nodal ligand provided.

We thank the reviewer for this suggestion and have co-expressed *vg1-sfGFP* with different levels of *cyc* and *sqt*. There is a notable amount of variation in the degree of extracellular localization of Vg1-sfGFP in combination with any given concentration of Nodal, so we have not drawn any quantitative conclusions about the effect of changing the ratio of Vg1:Nodal on Vg1 secretion. However, Vg1 secretion is no longer detectable or only detectable in a small number of cells when co-expressed with *cyc* or *sqt* at 40x lower concentration. We have quantified the number of embryos with intracellular versus extracellular puncta or diffuse protein localization (Figure 4—figure supplement 1, Table 1).

b) The distribution of Sqt-dsRED in Figure 4 looks very different from that of Sqt-GFP in Figure 2 and from that of Vg-1 in the presence of Sqt in Figure 2. Why is this so?

We thank the reviewer for this careful observation. We have found that co-expression of *vg1-sfGFP* with *sqt* can result in two different localization patterns: extracellular diffuse, and extracellular puncta. In light of reviewer 2’s observation we have made this statement more clear in the text, quantified each condition (Table 1), and provided an example of Vg1-sfGFP/Sqt-RFP diffuse co-localization (Figure 4).

3) If Nodal signaling is mediated by heterodimers of Nodal and Vg1, the extent of Nodal signaling should depend on their stoichiometry. It would be interesting to overexpress different relative amounts of Vg1 and Nodal ligand in Vg1 mutant and analyse Nodal signaling activation (via Smad2 nuclear localization for instance).

We thank the reviewer for this suggestion. Please see our last response to point 1 above.

4) The authors provide a simple theoretical model to show how the presence of maternally contributed Vg1 speeds up Nodal signaling upon production of Nodal ligands. I feel this is not particularly revealing: the idea that one component of a pathway being readily available speeds up signaling does not require modeling to be put forward. Moreover, the model is based on parameters measured or inferred for the Nodal ligands, which do not necessarily hold true for Vg1.

We agree with reviewer 2 that our embryonic model is intuitive and doesn’t necessarily require a mathematical model to make the point. However, after speaking with a number of colleagues, we found that the idea was not intuitive to everyone, so we decided to present a simple mathematical model.

Reviewer #3:[…] 1) Do the indel vg1 mutations lead to reduction of maternal vg1 message?

*in situ* hybridizations indicate that the maternal message is not visibly reduced in the mutant embryos, but injection of the mRNA containing the 8 bp deletion found in the mutant fish does not rescue the phenotype. This data is now included in Figure 1.

2) Are zygotic vg1 mutant truly aphenotypic? What about LR symmetry?

We thank the reviewer for raising this point. We have included data that *southpaw* expression and heart positioning in zygotic *vg1* mutants appears largely normal, suggesting that zygotic *vg1* is not absolutely required for left-right asymmetry (Figure 1—figure supplement 2).

3) Based on the ability of 50 pg of RNA encoding Vg1-NC not to rescue Mvg1 mutant phenotype the authors conclude that endogenous cleavage is required for Vg1 function appears too strong. Higher RNA doses should be tested.

We thank the reviewer for raising this point. We have tested up to 200 pg of *vg1-NC* mRNA, and find it does not rescue the phenotype (now indicated in the text).

4) Immunoblot experiments presented in Figure 3 should be quantified; there are no loading controls in these experiments.

We thank the reviewer for her careful analysis of the immunoblots. We agree that loading controls are necessary to draw quantitative conclusions about western blots, however, in this figure we only conclude that Vg1 is cleaved in the presence of Nodal, whereas Vg1-NC is not. The band corresponding to full-length Vg1/Vg1-NC provides an indication of the amount of Vg1 protein in each lane. We therefore don’t believe that a loading control is necessary in this case.

5) Experiments testing the intracellular localization of Vg1(Cyc;Sqt)-sfGFP would be more compelling if RNAs encoding these molecules would be injected with RNA encoding membraneRFP.

We thank the reviewer for this comment. In lieu of membraneRFP, we used Vg1-pHluorin in Figure 4 to verify that the Vg1 puncta we see in the presence of Nodal are extracellular. Moreover, the Nodal localization patterns have been previously published and validated (Muller et al. Science2012).

6) What RNA doses were injected in embryos shown in Figure 4?

The RNA doses are stated in the figure legend: 50 pg for each mRNA.

7) Experiments presented in Figure 4, would be more informative if sqt (bmp) RNAs were co-injected with a lineage tracer. There are two cells in the sqt panel showing intracellular Vg1-sfGFP, likely these cells had little/no Sqt, what would further strengthen the authors' conclusions.

We agree that there is variability in the extent of extracellular localization of Vg1 across individual embryos, which follows a general observation that mRNAs injected at the 1-cell stage are not uniformly inherited into subsequent cells. In order to address whether the extent of Vg1 secretion depends on the amount of Nodal, we co-expressed *vg1-sfGFP* with different levels of *cyc* and *sqt* and found that extracellular Vg1 is reduced when Nodallevels are lowered (see Figure 4—figure supplement 1). In addition, we have expressed *vg1-sfGFP* at the 1-cell stage and injected 1 cell out of 16 at the 16-cell stage with *cyc-RFP*. Only the cells that inherit Cyc-RFP secrete Vg1-sfGFP (Figure 4).

8) "…we performed co-immunoprecipitation experiments and found that Vg1 co-immunoprecipitated with Cyc or Sqt". This sentence could be understood that endogenous proteins were analyzed. Key experimental information is missing about how the relevant tagged proteins were expressed, at what stage of development co-IPs were performed. The co-IP with Cyc-HA is not convincing.

We thank the reviewer for bringing this to our attention. We have clarified the text, and included an additional co-IP with Vg1-HA and Cyc-/Sqt-Flag. We find that full-length Cyc protein readily co-precipitates with Vg1, but the mature Cyc protein is less easily detected than mature Sqt.

9) The conclusion that co-expression of Vg1 and Nodal in the same cells is required for activity, based on the experiments presented in Figure 5 is intriguing, especially for injection of vg1 RNA into the YSL of Mvg1 mutants. Whereas it has been previously shown that Nodal expression in YSL is sufficient to induce mesoderm and endoderm in the overlying blastoderm, this experiment would indicate that Cyc-Vg1 and Sqt-Vg1 heterodimers secreted from YSL are sufficient to create Nodal activity gradient to not only induce but also pattern the blastoderm. However, this experiment raises many questions: has injection been truly confined to the YSL (there is no lineage tracing data presented and a sceptic could argue that the 13 embryos that were rescued express the ligands also in the blastoderm due to ineffective injection? The evidence that expression is confined to the YSL is essential. It would be also important to show that Smad2 activity in such embryos parallels WT pattern shown by the authors in their earlier publications. It would be also important to show that the ligands targeted by RNA injection to the YSL can be seen in extracellular spaces in the blastoderm.

We thank the reviewer for raising this interesting topic. Please see point 2 above.

10) How does the above result fit with the models of Nodal autoinduction and self-inhibition? In the model presented in Figure 7 and in the Discussion, the authors see an important role of Sqt and Cyc expressed at the embryonic margin not just in the YSL.

We thank the reviewer for raising this interesting point. Since Vg1 and Nodal are expressed in both the YSL and embryonic margin, it is likely that Vg1-Nodal heterodimers normally form in both locations. The reviewer is correct to point out, however, that the YSL-mediated rescue suggests that the formation of heterodimers in the YSL alone might be sufficient for patterning, and supports the view that Nodal signaling can be propagated through diffusion, thus undermining some of the claims in (van Boxtel et al. Dev. Cell 2015). Since the rescue is not fully penetrant, we are currently performing complementary experiments to directly test different models of Nodal propagation and will report these results in a separate study.